# Toward a Stable, Fair, and Comprehensive Evaluation of Object Hallucination in Large Vision-Language Models

**Hongliang Wei, Xingtao Wang**[*]**, Xianqi Zhang, Xiaopeng Fan, Debin Zhao**
Harbin Institute of Technology
Harbin, China

## Abstract

Given different instructions, large vision-language models (LVLMs) exhibit different degrees of object hallucinations, posing a significant challenge to the evaluation of object hallucinations. Overcoming this challenge, existing object hallucination evaluation methods average the results obtained from a set of instructions. However, these methods fail to provide consistent evaluation across instruction sets that generate image descriptions of significantly different lengths. In this paper, we present the first systematic investigation into the effect of instructions on object hallucinations in LVLMs, with a specific focus on the role played by image description lengths. A valuable finding is that instructions indirectly affect hallucinations through the length of image descriptions. The longer the image description, the higher the object hallucination degree. Accordingly, we fit an informative length-hallucination curve, upon which a fine-grained evaluation framework named LeHaCE is introduced for evaluating object hallucinations at any given image description length. LeHaCE evaluates the object hallucination degree at a uniform image description length to mitigate the effect of description lengths, promoting stability and fairness. Moreover, LeHaCE incorporates the curve slope as an innovative hallucination evaluation metric, reflecting the extent to which the object hallucination degree is affected by the image description length, achieving a more comprehensive evaluation. Experimental results demonstrate that LeHaCE provides a more stable, fair, and comprehensive evaluation of object hallucinations in LVLMs compared to existing methods.

## 1 Introduction

Drawing inspiration from the remarkable language capabilities exhibited by large language models (LLMs) [1–3], large vision-language models (LVLMs) [2, 4–7] have been well-developed, achieving significant advancements in complex multimodal tasks. However, the practical application of LVLMs is heavily hindered by hallucination phenomena [8, 9], which refer to situations where objects in image descriptions generated by LVLMs are inconsistent with the provided visual content. Considerable efforts have been dedicated to both evaluation [9–11] and mitigation [12–14] of hallucination phenomena, leading to notable advancements.

A significant challenge in object hallucination evaluation arises from the effect of instructions on object hallucinations [9]. Overcoming this challenge, existing object hallucination evaluation methods typically adopt an average-based framework, which averages the results obtained from a set of instructions. However, as shown in Figure 1, this framework fails to provide consistent evaluation across instruction sets that generate image descriptions of significantly varying lengths. Specifically,

---

[*]Corresponding author.

38th Conference on Neural Information Processing Systems (NeurIPS 2024).

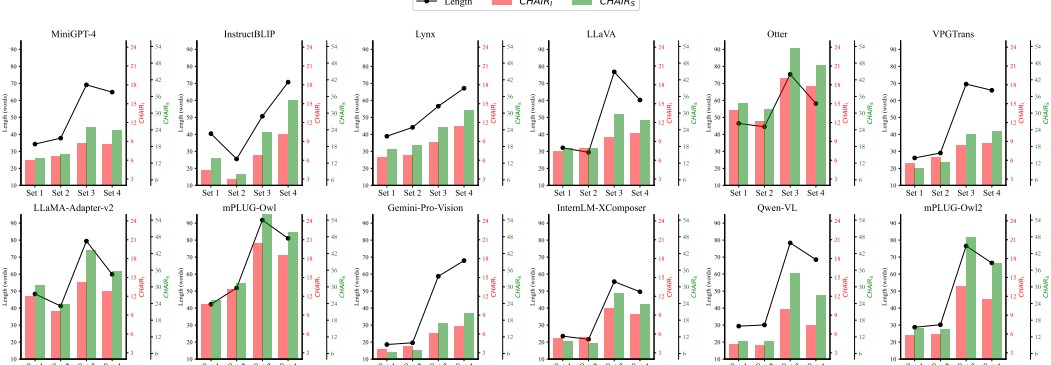

Figure 1: The evaluation results of LVLMs on four instruction sets using the CHAIR with the average-based framework. Length refers to the average length of generated image descriptions. Each instruction set consists of six distinct instructions, and there is no overlap between instructions in different sets. All instructions prompt LVLMs to describe the image.

while evaluation results of LVLMs remain consistent across certain instruction sets (e.g., set 1 and set 2), inconsistencies arise when comparing instruction sets with significantly different average image description lengths (e.g., set 2 and set 3).

In this paper, we present the first systematic investigation into the effect of instructions on object hallucinations in LVLMs, with a specific focus on the role played by the length of image descriptions. Technically, we evaluate lengths and object hallucination degrees (measured by CHAIR scores) of the image descriptions generated by LVLMs under different instructions (see Section 3.1 for more details). The experimental results are shown in Figures 2 & 3, from which we can observe that the degree of object hallucination is primarily influenced by the length of image descriptions, with instructions only indirectly affecting hallucinations through their effect on description lengths. The longer the image description, the higher the object hallucination degree, and there is a clear linear relation between them. Hence, it is imperative to take into account the length of image descriptions in hallucination evaluation. Unfortunately, the average-based framework can only select instructions, without the ability to directly control the length of image descriptions.

Motivated by the findings, we propose a fine-grained evaluation framework called LeHaCE, which fits an informative length-hallucination curve to evaluate object hallucinations at any given image description length within a large range. LeHaCE evaluates the object hallucination degree at a uniform image description length to mitigate the effect of image description length, ensuring stable evaluations for the same LVLM across different instruction sets and fair comparisons among different LVLMs. Moreover, LeHaCE incorporates the curve slope as an innovative hallucination evaluation metric, reflecting the extent to which the object hallucination degree is influenced by the image description length, achieving a more comprehensive evaluation. Experiment results on 12 representative LVLMs show that LeHaCE can evaluate object hallucinations of LVLMs in a more stable, fair, and comprehensive way.

The main contributions of this paper are summarized as follows:

- We conduct the first systematic investigation into the effect of instructions on object hallucinations in LVLMs and find that the degree of object hallucinations is primarily influenced by the length of image descriptions, with instructions only indirectly affecting hallucinations through their effect on image description lengths.

- We propose an object hallucination evaluation framework called LeHaCE, which fits an informative length-hallucination curve to evaluate object hallucination at a uniform image description length, realizing a more stable and fair evaluation.

- We employ the curve slope as an innovative hallucination evaluation metric, reflecting the extent to which the object hallucination degree is affected by the image description length, achieving a more comprehensive evaluation.

## 2 Related work

### 2.1 Large Vision-Language Models

Inspired by the success of LLMs in NLP [1–3], researchers have extended LLMs to multimodal tasks [15–28], proposing numerous LVLMs and achieving new advancements [7, 14, 29–35]. These LVLMs align the multi-modal encoders with LLM through multitask fine-tuning and instruction fine-tuning on multi-modal datasets, enabling LLM to acquire multi-modal perception and instruction-following capabilities. Specifically, to integrate multimodal features, Flamingo [29] proposes a cross-attention structure to achieve arbitrary interleaved multi-modal feature fusion. BLIP-2 [35] introduces Q-Former to bridge the visual backbone model and LLM. mPLUG-Owl2 [36] introduces a modality adaptive module to facilitate the fusion between different modules. To enhance generalization and improve instruction-following capabilities, some methods [4–6, 12, 37] propose multi-task fine-tuning and instruction fine-tuning for LVLMs. Among them, LRV-instruction [12], MiniGPT-4 [5], LLaVA [6] and SViT [37] employ ChatGPT to augment instruction data. To mitigate the risk of catastrophic forgetting of language knowledge during the training process, mPLUG-Owl [38] and LLaVA-1.5 [6] perform joint training on pure language and visual-language instructional data. More recently, mPLUG-DocOwl [31], InternLM-XComposer [32], Kosmos-2 [33], Shikra [34], Cantor [39], BuboGPT [30], and Qwen-VL [7] further enhance the capabilities of LVLMs in optical character recognition, document understanding, multi-modal interleaved composition and visual grounding.

### 2.2 Hallucination in LVLMs

Works on the hallucination in LVLMs focus on two aspects: evaluation and mitigation. For the hallucination evaluation, POPE [9] designs a polling-based query method to avoid the influence of instructions on hallucination evaluation. By presenting LVLMs with brief "yes" or "no" questions regarding the target of detection, the evaluation of hallucination is transformed into a simple binary classification task. NOPE [10] designs a novel benchmark to evaluate the performance of LVLMs in recognizing the non-existence of objects in visual questions. AMBER [11] designs a multi-dimensional LVLMs hallucination evaluation benchmark without LLMs, targeting existence, attribute, and relation hallucination. For the hallucination mitigation, LRV-Instruction [40] creates a balanced set of positive and negative instructions to perform robust visual instruction adjustment for LVLMs. VIGC [14] employs an iterative approach to generate detailed and accurate answers gradually. Woodpecker [13] proposes a post-processing method that utilizes expert models to locate and correct hallucinations from generated text.

While existing methods [9] observe that the hallucination degree of LVLMs is unstable across different instructions, this phenomenon has not been thoroughly investigated to date. This work presents the first comprehensive study investigating the influence of instructions on the hallucination rate of LVLMs. Building upon our findings, we propose LeHaCE framework, which can evaluate hallucination of LVLMs in a more stable and comprehensive manner. Contrary to polling-based query methods, LeHaCE can directly evaluate the hallucination rate of image descriptions generated by LVLMs, which is more in line with the practical application scenarios of LVLMs.

## 3 Hallucination of LVLMs Under Different Instructions

This section provides the investigation into the effect of instructions on hallucinations, with a specific focus on the role played by the length of image descriptions. The experimental settings are presented initially, followed by a comprehensive analysis of the experimental results

### 3.1 Experimental Settings

In this investigation, twelve popular LVLMs are included, namely Gemini-Pro-Vision pro [2], Qwen-VL [7], MiniGPT-4 [5], LLaVA [6], InstructBLIP [4], LLaMA-Adapter-v2 [41], mPLUG-Owl2 [36], mPLUG-Owl [38], InternLM-XComposer [32], VPGTrans [42], Otter [43] and Lynx [44]. All LVLMs are prompted by 25 different instructions to generate image descriptions for 256 images in MSCOCO [45]. All descriptions are generated using beam search with a beam size of 5. For the instructions, we utilize those from [4] and additionally propose several others, as detailed in the appendix. We use CHAIR [8] as the evaluation metric for hallucinations, which has two variants: $CHAIR_I$ and $CHAIR_S$. Given the ground truth objects in the image, $CHAIR_I$ calculates the proportion

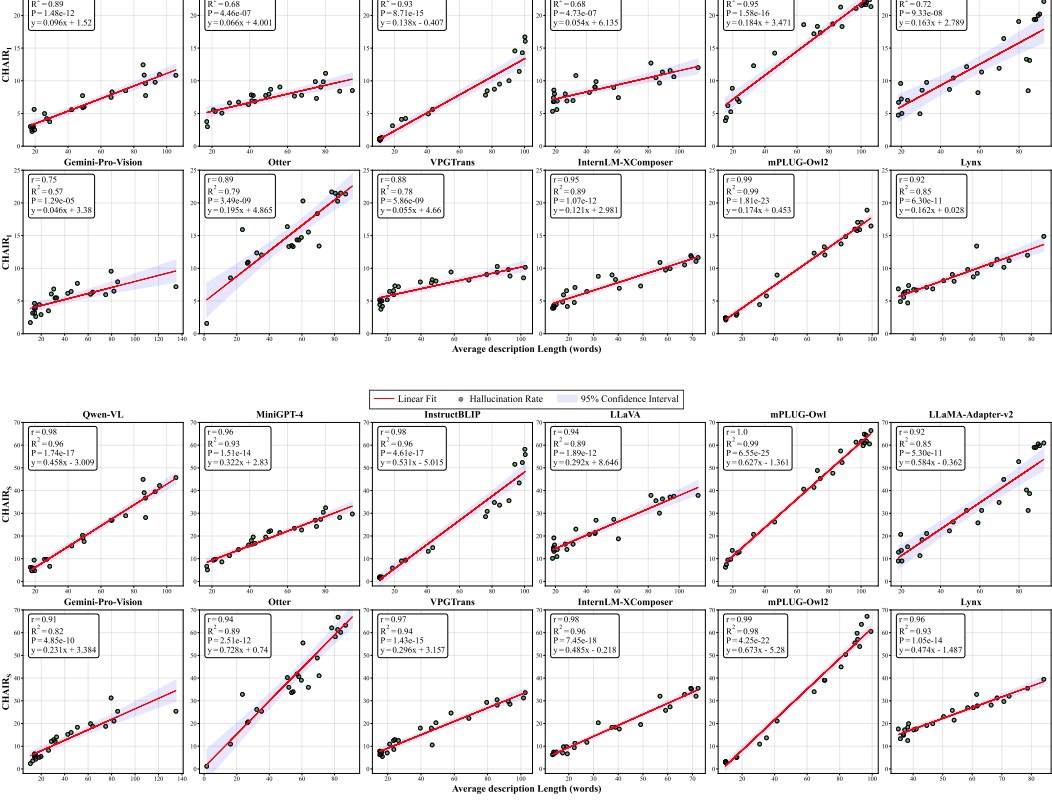

Figure 2: Scatter plots of CHAIR scores and average lengths of the 25 sets of image descriptions generated by 25 instructions. $r$ denotes the Pearson correlation coefficient between the hallucination rates and the average image description lengths, $R^2$ and $P$ represent the coefficient of determination and $p$-value respectively for the linear regression.

of objects that appear in the descriptions but not in the image, while CHAIR$_S$ is the proportion of descriptions that include hallucination. Formally, CHAIR$_I$ and CHAIR$_S$ can be expressed as follows:

$$\text{CHAIR}_I = \frac{\left|\{\text{hallucinated objects}\}\right|}{\left|\{\text{all mentioned objects}\}\right|}, \tag{1}$$

$$\text{CHAIR}_S = \frac{\left|\{\text{descriptions with hallucinated objects}\}\right|}{\left|\{\text{all descriptions}\}\right|}. \tag{2}$$

For more experimental settings, we use the Pearson correlation coefficient to measure the correlation between the average length and the hallucination rate of image descriptions. Lengths are measured in word count.

### 3.2 Experimental Analysis

The results are presented in Figure 2 & 3, from which we get two key observations: **1)** Figure 2 shows the relationship between the hallucination rate and the average image description length, we can observe that the hallucination rate increases with the average image description length and there is a clear linear correlation between them. Specifically, the Pearson correlation coefficient between hallucination rates and the average image description lengths exceeds 0.6 for all LVLMs, with 10 LVLMs exceeding 0.8 and 5 LVLMs exceeding 0.9. **2)** Figure 3 shows the impact of instructions on the length of image descriptions generated by LVLMs, from which we can observe that the length of image descriptions generated by the same LVLM with

different instructions can vary significantly, e.g., Gemini-Pro-Vision with Instruction 11 (101 words in average) v.s. Gemini-Pro-Vision with Instruction 18 (20 words in average). Furthermore, the length of image descriptions generated by different LVLMs with the same instruction can also differ greatly, e.g., MiniGPT-4 (11 words in average) v.s. Gemini-Pro-Vision (97 words in average) with Instruction 17.

Based on the aforementioned two observations, we can draw the following conclusions: **1)** The degree of object hallucinations is primarily influenced by the length of image descriptions, with instructions only indirectly affecting hallucinations through their effect on image description lengths. Hence, it is imperative to take into account the length of image descriptions in hallucination evaluation. However, controlling the length of image descriptions generated by LVLMs is challenging [2], given that even subtle semantic differences between instructions can significantly impact the output length of LVLMs (shown in Figure 3). **2)** In addition to the hallucination degree , the rate at

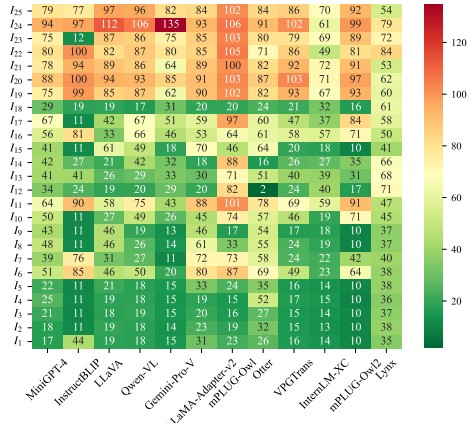

Figure 3: The average lengths of image descriptions generated by LVLMs when prompted by different instructions.

which hallucination degree increase with description length is also a meaningful indicator for characterizing the nature of LVLMs hallucinations. Considering both the hallucination degree and the growth rate of hallucination degree can provide a more comprehensive evaluation for hallucinations in LVLMs. For example, as shown in the Figure 2, although InstructBLIP has the lowest hallucination degrees in short image descriptions, it exhibits high instability with a rapid increase in hallucination degrees, resulting in high hallucination in long image descriptions.

## 4 Length-Hallucination Curve Based Hallucination Evaluation Framework

In this section, we first introduce the average-based hallucination evaluation framework and discuss its limitations. Then, we elaborate on the proposed LeHaCE framework and evaluate representative LVLMs with LeHaCE. Finally, the stability of LeHaCE is analysed.

### 4.1 Average-Based Hallucination Evaluation Framework

The average-based hallucination evaluation framework mitigates the challenge caused by instructions by averaging the hallucination rates over different instructions. Formally, the hallucination rates and average lengths of the image descriptions generated by the LVLM under N instructions are denoted as $\{\ell_i, hr_i\}_{i=1}^{N}$. The average hallucination rate $\bar{hr}$ and average length $\bar{\ell}$ of image descriptions over all instructions can be calculated as follows: $\bar{hr} = \frac{1}{N}\sum_{i=1}^{N} hr_i$ and $\bar{\ell} = \frac{1}{N}\sum_{i=1}^{N} \ell_i$. The average-based hallucination evaluation framework utilizes $\bar{hr}$ to evaluate the hallucination of LVLMs.

However, due to substantial variations in the average lengths of image descriptions generated by different instruction sets, the average-based framework struggles to mitigate the effect of image description lengths on object hallucinations, resulting in unstable and unfair evaluations. Specifically, as shown in Figure 4 (left), when the average-based framework evaluates an LVLM under different instruction sets, the inconsistent average image description lengths lead to unstable evaluation. Moreover, Figure 4 (right) shows that when the average-based framework evaluates different LVLMs under the same instructions, the inconsistent average image description lengths lead to unfair evaluation.

### 4.2 Length-Hallucination Curve Based Hallucination Evaluation Framework

Section 3.2 reveals the significant effect of image description lengths on the hallucination degree. To mitigate this effect, it is crucial to control the description length during the hallucination evaluation.

---

[2]This work does not consider controlling length by truncating the generated descriptions, only considering cases where LVLMs generate complete image descriptions, as this better fits practical application scenarios.

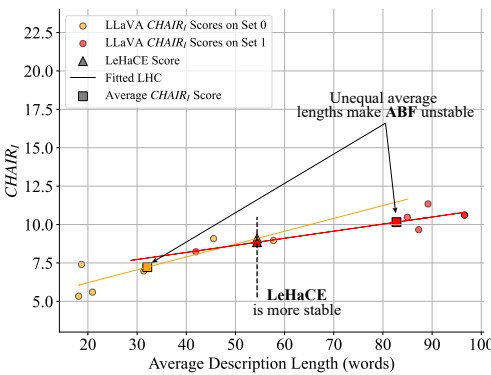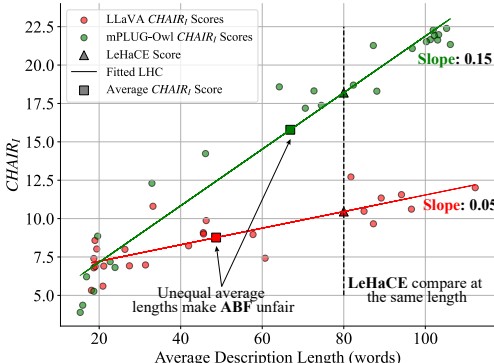

Figure 4: **Illustrations** of the average-based evaluation framework (ABF) and our LeHaCE framework. The **left** figure presents the object hallucination evaluation of LLaVa on two instruction sets. The **right** figure presents the object hallucination evaluation of LLaVa and mPLUG-Owl on the same set of instructions.

However, controlling the length of generated image descriptions is challenging because LVLMs are highly sensitive to instructions. To address this challenge, we fit a length-hallucination curve to evaluate LVLMs at any desired length. Specifically, based on the clear linear correlation observed in Section 3.2, we assume a linear correlation between image description lengths and hallucination rates of LVLMs. Figure 4 intuitively illustrates the LeHaCE framework.

Formally, we use $\{\ell_i, hr_i\}_{i=1}^{N}$ to represent the average lengths and hallucination rates of image descriptions generated by the LVLM under N instructions. The linear regression curve of $\{\ell_i, hr_i\}_{i=1}^{N}$, which we refer to as the Length-Hallucination Curve (LHC), can be formalized as follows:

$$\text{LHC}\left(\ell\right) = \beta * \ell + \alpha, \tag{3}$$

where $\beta$ and $\alpha$ are:

$$\beta = \frac{\sum_{i=1}^{N}\left(\ell_i - \bar{\ell}\right)\left(hr_i - \bar{hr}\right)}{\sum_{i=1}^{N}\left(\ell_i - \bar{\ell}\right)^2}, \tag{4}$$

$$\alpha = \bar{hr} - \left(\beta * \bar{\ell}\right). \tag{5}$$

Length-hallucination curve summarizes the trend between the hallucination rate and the description length. The regression coefficient $\beta$ represents the rate at which the hallucination rate increases with the growth of description length. LeHaCE uses the LHC to evaluate the hallucination in LVLMs. LeHaCE consists of two metrics:

$$\text{LeHaCE}\left(\ell\right) = \text{LHC}\left(\ell\right), \tag{6}$$

$$\text{LeHaCE}_{GR} = \beta \tag{7}$$

where $\text{LeHaCE}\left(\ell\right)$ measures the hallucination rate at the specified length $\ell$, and $\text{LeHaCE}_{GR}$ measures the rate at which the hallucination rate increases with the increase in description length. Note that LeHaCE can be built upon any hallucination degree evaluation metric, enhancing their stability, fairness, and comprehensiveness. In this paper, we use CHAIR as the metric for measuring the hallucination degree.

Compared to the average-based hallucination evaluation framework, LeHaCE has three advantages. As intuitively shown in Figure 4, LeHaCE can evaluate the hallucination degree of LVLMs at a uniform image description length, thereby mitigating the influence of description length on hallucination degree and resulting in a **more stable and fair** evaluation. Moreover, LeHaCE can evaluate the hallucination degree at multiple lengths and the growth rate of hallucination degree, leading to a **more comprehensive** evaluation.

## 4.3 Evaluation on MSCOCO and NoCaps

We evaluate twelve LVLMs with LeHaCE at lengths of 20, 40, 60, and 80 words. This evaluation is conducted on subsets of the MSCOCO [45] test set and the NoCaps [46] validation set, each compris-

| Model | $L_{C_I}(20)$ | $L_{C_I}(40)$ | $L_{C_I}(60)$ | $L_{C_I}(80)$ | $L_{C_I}$GR | $L_{C_S}(20)$ | $L_{C_S}(40)$ | $L_{C_S}(60)$ | $L_{C_S}(80)$ | $L_{C_S}$GR |
|---|---|---|---|---|---|---|---|---|---|---|
| | | | | | **MSCOCO** | | | | | |
| MiniGPT-4 | 5.33 | 6.66 | 7.98 | 9.31 | 0.07 | 9.27 | 15.71 | 22.15 | 28.59 | 0.32 |
| InstructBLIP | **2.35** | **5.10** | 7.86 | 10.61 | 0.14 | **5.61** | 16.24 | 26.87 | 37.50 | 0.53 |
| Lynx | 3.26 | 6.49 | 9.72 | 12.95 | 0.16 | 8.00 | 17.48 | 26.97 | 36.46 | 0.47 |
| LLaVA | 7.22 | 8.30 | 9.38 | 10.46 | **0.05** | 14.48 | 20.31 | 26.14 | 31.97 | 0.29 |
| Otter | 8.76 | 12.66 | 16.56 | 20.45 | 0.19 | 15.31 | 29.88 | 44.45 | 59.02 | 0.73 |
| VPGTrans | 5.77 | 6.87 | 7.97 | 9.08 | 0.06 | 9.08 | 15.01 | 20.94 | 26.86 | 0.30 |
| LLaMA-Adapter-v2 | 6.04 | 9.29 | 12.54 | 15.80 | 0.16 | 11.31 | 22.99 | 34.66 | 46.34 | 0.58 |
| mPLUG-Owl | 7.15 | 10.84 | 14.52 | 18.20 | 0.18 | 11.18 | 23.71 | 36.25 | 48.79 | 0.63 |
| Gemini-Pro-Vision | 4.30 | 5.22 | **6.15** | **7.07** | **0.05** | 8.00 | **12.61** | **17.22** | **21.83** | **0.23** |
| InternLM-XComposer | 5.40 | 7.82 | 10.25 | 12.67 | 0.12 | 9.48 | 19.18 | 28.88 | 38.58 | 0.48 |
| Qwen-VL | 3.44 | 5.36 | 7.28 | 9.20 | 0.10 | 6.15 | 15.31 | 24.47 | 33.63 | 0.46 |
| mPLUG-Owl2 | 3.92 | 7.39 | 10.86 | 14.33 | 0.17 | 8.19 | 21.66 | 35.12 | 48.59 | 0.67 |
| | | | | | **NoCaps** | | | | | |
| MiniGPT-4 | 14.53 | 16.79 | 19.05 | 21.30 | 0.11 | 23.75 | 35.75 | 47.76 | 59.77 | 0.60 |
| InstructBLIP | **6.52** | **10.20** | 13.88 | 17.56 | 0.18 | 13.33 | 26.39 | 39.45 | 52.50 | 0.65 |
| Lynx | 13.79 | 17.18 | 20.57 | 23.96 | 0.17 | 36.07 | 46.11 | 56.16 | 66.21 | 0.50 |
| LLaVA | 12.68 | 14.48 | 16.29 | 18.09 | **0.09** | 24.15 | 33.90 | 43.66 | 53.42 | **0.49** |
| Otter | 15.49 | 19.03 | 22.58 | 26.12 | 0.18 | 25.38 | 38.89 | 52.40 | 65.91 | 0.68 |
| VPGTrans | 12.51 | 14.39 | 16.26 | 18.14 | **0.09** | 20.39 | 31.95 | 43.51 | 55.07 | 0.58 |
| LLaMA-Adapter-v2 | 12.52 | 16.07 | 19.62 | 23.17 | 0.18 | 22.44 | 35.31 | 48.18 | 61.04 | 0.64 |
| mPLUG-Owl | 12.85 | 15.84 | 18.84 | 21.83 | 0.15 | 19.77 | 30.68 | 41.60 | 52.52 | 0.55 |
| Gemini-Pro-Vision | 12.76 | 15.17 | 17.57 | 19.98 | 0.12 | 22.63 | 34.56 | 46.50 | 58.44 | 0.60 |
| InternLM-XComposer | 10.93 | 12.74 | 14.54 | 16.34 | **0.09** | 20.12 | 31.22 | 42.33 | 53.44 | 0.56 |
| Qwen-VL | 8.37 | 10.69 | **13.01** | **15.33** | 0.12 | 14.15 | **25.00** | **35.85** | **46.71** | 0.54 |
| mPLUG-Owl2 | 6.91 | 10.82 | 14.72 | 18.63 | 0.20 | **11.72** | 25.45 | 39.17 | 52.90 | 0.69 |

Table 1: LeHaCE scores of LVLMs on the MSCOCO and NoCaps datasets. $L_{C_I}$ and $L_{C_S}$ represent CHAIR$_I$ and CHAIR$_S$ with the LeHaCE framework. The best result on each metric for each dataset is represented in bold, and the second best result is indicated with an underline.

ing randomly selected 256 images. The length-hallucination curve in LeHaCE is fitted on the CHAIR scores of image descriptions generated by 25 different instructions. To calculate CHAIR scores on No-Caps, we follow the setting proposed in [8, 47]. All descriptions are generated using beam search with a beam size of 5. The experiments are conducted with PyTorch on Nvidia GeForce RTX 3090 GPUs.

The results are shown in Table 1, which demonstrate that LeHaCE can evaluate the object hallucination degree of LVLMs at given image description lengths, as well as the growth rate of the hallucination degree, providing a fair and comprehensive evaluation. Specifically, **1)** For short descriptions, InstructBLIP achieves the best performance on both the MSCOCO and NoCaps datasets. However, its higher growth rate of hallucination degree leads to poor performance on longer descriptions. **2)** For medium-length and long descriptions, Gemini-Pro-Vision and Qwen-VL exhibit the best performance on the MSCOCO and NoCaps datasets, respectively. This is attributed to their relatively small growth rate in hallucination degree. **3)** Gemini-Pro-Vision and LLaVA exhibit the lowest growth rate in hallucination degree on the MSCOCO and NoCaps datasets, respectively.

In Table 1, LVLMs exhibit higher degrees of hallucination on the NoCaps dataset compared to the MSCOCO dataset. This is attributed to the fact that LVLMs typically use the MSCOCO for training, making the NoCaps dataset an out-of-distribution dataset. The results show that the distributional differences not only increase the hallucination degree of LVLMs at various description lengths but also amplify the growth rate of hallucination degree.

## 4.4 Stability of LeHaCE

As mentioned above, LeHaCE evaluates the hallucination degree of LVLMs in a more stable manner. This subsection verifies the stability of the proposed LeHaCE framework. Specifically, LVLMs are prompted by three sets of different instructions to generate three sets of image descriptions. Each instruction set consists of multiple instructions randomly drawn from a pool of 25 instructions, with no overlap between instructions in different sets. The image descriptions generated by different instructions in each set are evaluated using the LeHaCE framework and the average-based framework, respectively. The stability of the LeHaCE and the average-based frameworks on the three sets of image descriptions is evaluated using the **Relative Standard Deviation (RSD)**, which is defined as the ratio of the standard deviation $\sigma$ to the mean $\mu$, $RDS = \sigma/\mu$. The lower the RSD, the more stable

| # of Ins | Gemini-Pro-Vision | | | | Qwen-VL | | | | MiniGPT-4 | | | | LLaVA | | | |
|---|---|---|---|---|---|---|---|---|---|---|---|---|---|---|---|---|
| | $C_I$ | $L_{C_I}$ | $C_S$ | $L_{C_S}$ | $C_I$ | $L_{C_I}$ | $C_S$ | $L_{C_S}$ | $C_I$ | $L_{C_I}$ | $C_S$ | $L_{C_S}$ | $C_I$ | $L_{C_I}$ | $C_S$ | $L_{C_S}$ |
| 3 | **0.16** | 0.25 | 0.32 | **0.18** | 0.29 | **0.11** | 0.41 | **0.09** | 0.14 | **0.09** | 0.22 | **0.07** | **0.16** | 0.23 | 0.29 | **0.23** |
| 4 | **0.14** | 0.16 | 0.30 | **0.13** | 0.27 | **0.11** | 0.38 | **0.15** | 0.12 | **0.08** | 0.21 | **0.06** | **0.13** | 0.20 | 0.23 | **0.18** |
| 5 | 0.16 | **0.15** | 0.27 | **0.12** | 0.22 | **0.08** | 0.34 | **0.05** | 0.08 | **0.06** | 0.15 | **0.05** | 0.10 | **0.08** | 0.20 | **0.08** |
| 6 | 0.13 | **0.09** | 0.24 | **0.10** | 0.21 | **0.07** | 0.32 | **0.06** | 0.10 | **0.04** | 0.16 | **0.04** | 0.10 | **0.07** | 0.19 | **0.06** |
| 7 | 0.15 | **0.12** | 0.22 | **0.11** | 0.18 | **0.06** | 0.25 | **0.05** | 0.10 | **0.06** | 0.15 | **0.06** | 0.10 | **0.06** | 0.18 | **0.07** |
| 8 | 0.12 | **0.09** | 0.20 | **0.08** | 0.15 | **0.06** | 0.22 | **0.04** | 0.06 | **0.04** | 0.12 | **0.03** | 0.08 | **0.06** | 0.15 | **0.06** |

| # of Ins | InternLM | | | | Otter | | | | LLaMA-Adapter-v2 | | | | mPLUG-Owl | | | |
|---|---|---|---|---|---|---|---|---|---|---|---|---|---|---|---|---|
| | $C_I$ | $L_{C_I}$ | $C_S$ | $L_{C_S}$ | $C_I$ | $L_{C_I}$ | $C_S$ | $L_{C_S}$ | $C_I$ | $L_{C_I}$ | $C_S$ | $L_{C_S}$ | $C_I$ | $L_{C_I}$ | $C_S$ | $L_{C_S}$ |
| 3 | 0.19 | **0.13** | 0.31 | **0.13** | 0.24 | **0.13** | 0.31 | **0.14** | **0.27** | 0.75 | **0.32** | 0.80 | 0.19 | **0.08** | 0.25 | **0.06** |
| 4 | 0.18 | **0.07** | 0.27 | **0.06** | 0.19 | **0.07** | 0.24 | **0.07** | 0.20 | **0.13** | 0.25 | **0.11** | 0.20 | **0.07** | 0.26 | **0.03** |
| 5 | 0.17 | **0.08** | 0.28 | **0.08** | 0.15 | **0.08** | 0.20 | **0.09** | 0.22 | **0.10** | 0.27 | **0.09** | 0.16 | **0.06** | 0.23 | **0.02** |
| 6 | 0.16 | **0.05** | 0.27 | **0.06** | 0.18 | **0.09** | 0.23 | **0.09** | 0.17 | **0.10** | 0.21 | **0.09** | 0.14 | **0.04** | 0.18 | **0.02** |
| 7 | 0.15 | **0.05** | 0.22 | **0.05** | 0.14 | **0.05** | 0.19 | **0.05** | 0.15 | **0.08** | 0.19 | **0.07** | 0.14 | **0.04** | 0.17 | **0.02** |
| 8 | 0.09 | **0.05** | 0.16 | **0.05** | 0.10 | **0.06** | 0.14 | **0.06** | 0.17 | **0.08** | 0.21 | **0.07** | 0.15 | **0.04** | 0.20 | **0.02** |

| # of Ins | InstructBLIP | | | | mPLUG-Owl2 | | | | Lynx | | | | VPGTrans | | | |
|---|---|---|---|---|---|---|---|---|---|---|---|---|---|---|---|---|
| | $C_I$ | $L_{C_I}$ | $C_S$ | $L_{C_S}$ | $C_I$ | $L_{C_I}$ | $C_S$ | $L_{C_S}$ | $C_I$ | $L_{C_I}$ | $C_S$ | $L_{C_S}$ | $C_I$ | $L_{C_I}$ | $C_S$ | $L_{C_S}$ |
| 3 | 0.43 | **0.28** | 0.53 | **0.20** | 0.39 | **0.34** | 0.50 | **0.27** | **0.13** | 0.19 | **0.14** | **0.14** | **0.14** | 0.17 | 0.32 | **0.14** |
| 4 | 0.37 | **0.12** | 0.44 | **0.10** | 0.32 | **0.05** | 0.41 | **0.09** | 0.13 | **0.07** | 0.14 | **0.06** | **0.13** | 0.14 | 0.30 | **0.13** |
| 5 | 0.43 | **0.15** | 0.50 | **0.11** | 0.33 | **0.04** | 0.44 | **0.06** | 0.12 | **0.07** | 0.13 | **0.05** | **0.10** | 0.10 | 0.26 | **0.09** |
| 6 | 0.28 | **0.11** | 0.34 | **0.10** | 0.29 | **0.03** | 0.37 | **0.04** | 0.10 | **0.05** | 0.10 | **0.03** | 0.11 | **0.05** | 0.26 | **0.07** |
| 7 | 0.30 | **0.09** | 0.35 | **0.10** | 0.23 | **0.03** | 0.28 | **0.05** | 0.12 | **0.04** | 0.11 | **0.02** | 0.10 | **0.06** | 0.21 | **0.05** |
| 8 | 0.36 | **0.10** | 0.41 | **0.09** | 0.26 | **0.04** | 0.31 | **0.04** | 0.11 | **0.05** | 0.10 | **0.03** | 0.08 | **0.05** | 0.17 | **0.04** |

| # of Ins | Gemini-Pro-Vision | | | | Qwen-VL | | | | MiniGPT-4 | | | | LLaVA | | | |
|---|---|---|---|---|---|---|---|---|---|---|---|---|---|---|---|---|
| | $C_I$ | $L_{C_I}$ | $C_S$ | $L_{C_S}$ | $C_I$ | $L_{C_I}$ | $C_S$ | $L_{C_S}$ | $C_I$ | $L_{C_I}$ | $C_S$ | $L_{C_S}$ | $C_I$ | $L_{C_I}$ | $C_S$ | $L_{C_S}$ |
| 3 | **0.15** | 0.26 | 0.30 | **0.24** | **0.17** | 0.28 | 0.33 | **0.26** | 0.08 | **0.06** | 0.16 | **0.06** | 0.14 | **0.12** | 0.28 | **0.14** |
| 4 | 0.13 | **0.10** | 0.28 | **0.10** | 0.15 | **0.10** | 0.28 | **0.09** | 0.07 | **0.05** | 0.15 | **0.06** | 0.13 | **0.12** | 0.23 | **0.18** |
| 5 | 0.14 | **0.14** | 0.27 | **0.14** | 0.16 | **0.15** | 0.29 | **0.11** | 0.05 | **0.03** | 0.11 | **0.04** | 0.08 | **0.05** | 0.18 | **0.07** |
| 6 | 0.12 | **0.09** | 0.25 | **0.11** | 0.14 | **0.07** | 0.26 | **0.05** | 0.06 | **0.03** | 0.11 | **0.03** | 0.10 | **0.04** | 0.18 | **0.06** |
| 7 | 0.13 | **0.10** | 0.22 | **0.11** | 0.10 | **0.06** | 0.19 | **0.05** | 0.06 | **0.03** | 0.12 | **0.04** | 0.10 | **0.05** | 0.17 | **0.05** |
| 8 | 0.10 | **0.07** | 0.18 | **0.09** | 0.11 | **0.05** | 0.19 | **0.04** | 0.04 | **0.03** | 0.09 | **0.04** | 0.06 | **0.04** | 0.12 | **0.04** |

| # of Ins | InternLM | | | | Otter | | | | LLaMA-Adapter-v2 | | | | mPLUG-Owl | | | |
|---|---|---|---|---|---|---|---|---|---|---|---|---|---|---|---|---|
| | $C_I$ | $L_{C_I}$ | $C_S$ | $L_{C_S}$ | $C_I$ | $L_{C_I}$ | $C_S$ | $L_{C_S}$ | $C_I$ | $L_{C_I}$ | $C_S$ | $L_{C_S}$ | $C_I$ | $L_{C_I}$ | $C_S$ | $L_{C_S}$ |
| 3 | **0.12** | 0.58 | **0.26** | 0.44 | 0.16 | **0.06** | 0.25 | **0.09** | **0.17** | 0.33 | 0.25 | **0.23** | 0.14 | 0.15 | 0.22 | **0.10** |
| 4 | 0.09 | **0.07** | 0.22 | **0.07** | 0.12 | **0.05** | 0.19 | **0.05** | 0.13 | **0.06** | 0.20 | **0.05** | 0.14 | **0.06** | 0.23 | **0.05** |
| 5 | 0.08 | 0.08 | 0.21 | **0.05** | 0.10 | **0.04** | 0.15 | **0.05** | 0.13 | **0.04** | 0.19 | **0.04** | 0.11 | **0.05** | 0.18 | **0.04** |
| 6 | 0.09 | **0.05** | 0.20 | **0.03** | 0.12 | **0.05** | 0.18 | **0.05** | 0.11 | **0.04** | 0.16 | **0.03** | 0.09 | **0.04** | 0.15 | **0.03** |
| 7 | 0.07 | **0.04** | 0.16 | **0.03** | 0.10 | **0.04** | 0.14 | **0.04** | 0.10 | **0.05** | 0.14 | **0.03** | 0.09 | **0.04** | 0.15 | **0.03** |
| 8 | 0.05 | **0.04** | 0.13 | **0.03** | 0.06 | **0.04** | 0.10 | **0.04** | 0.11 | **0.05** | 0.14 | **0.03** | 0.10 | **0.03** | 0.16 | **0.03** |

| # of Ins | InstructBLIP | | | | mPLUG-Owl2 | | | | Lynx | | | | VPGTrans | | | |
|---|---|---|---|---|---|---|---|---|---|---|---|---|---|---|---|---|
| | $C_I$ | $L_{C_I}$ | $C_S$ | $L_{C_S}$ | $C_I$ | $L_{C_I}$ | $C_S$ | $L_{C_S}$ | $C_I$ | $L_{C_I}$ | $C_S$ | $L_{C_S}$ | $C_I$ | $L_{C_I}$ | $C_S$ | $L_{C_S}$ |
| 3 | 0.31 | **0.12** | 0.42 | **0.08** | 0.34 | **0.18** | 0.50 | **0.20** | 0.06 | **0.04** | 0.07 | **0.04** | **0.12** | 0.17 | 0.27 | **0.20** |
| 4 | 0.26 | **0.05** | 0.34 | **0.04** | 0.29 | **0.14** | 0.43 | **0.14** | 0.05 | **0.03** | 0.06 | **0.03** | **0.10** | 0.10 | 0.25 | **0.08** |
| 5 | 0.28 | **0.05** | 0.37 | **0.05** | 0.31 | **0.11** | 0.45 | **0.07** | 0.06 | **0.03** | 0.06 | **0.03** | 0.09 | **0.08** | 0.23 | **0.11** |
| 6 | 0.19 | **0.04** | 0.26 | **0.04** | 0.27 | **0.05** | 0.40 | **0.03** | 0.04 | **0.02** | 0.04 | **0.02** | 0.10 | **0.07** | 0.25 | **0.08** |
| 7 | 0.19 | **0.04** | 0.25 | **0.04** | 0.19 | **0.04** | 0.29 | **0.03** | 0.06 | **0.03** | 0.06 | **0.02** | 0.08 | **0.07** | 0.18 | **0.08** |
| 8 | 0.25 | **0.04** | 0.31 | **0.03** | 0.20 | **0.04** | 0.29 | **0.02** | 0.04 | **0.02** | 0.05 | **0.02** | 0.07 | **0.06** | 0.16 | **0.06** |

Table 2: The average RSD of CHAIR with the LeHaCE and the average-based frameworks, lower is better. $C_I$ and $C_S$ respectively represent CHAIR$_I$ and CHAIR$_S$ with the average-based hallucination evaluation framework. $L_{C_I}$ and $L_{C_S}$ respectively represent CHAIR$_I$ and CHAIR$_S$ with the LeHaCE framework. The best result under each setting is represented in bold.

| #Ins | MiniGPT-4 CHAIR$_S$ | | | CHAIR$_I$ | | | InstructBLIP CHAIR$_S$ | | | CHAIR$_I$ | | | LLaVA CHAIR$_S$ | | | CHAIR$_I$ | | |
|---|---|---|---|---|---|---|---|---|---|---|---|---|---|---|---|---|---|---|
| | L$_1$ | L$_2$ | L$_3$ | L$_1$ | L$_2$ | L$_3$ | L$_1$ | L$_2$ | L$_3$ | L$_1$ | L$_2$ | L$_3$ | L$_1$ | L$_2$ | L$_3$ | L$_1$ | L$_2$ | L$_3$ |
| 3 | **0.07** | 0.50 | 0.58 | **0.09** | 0.58 | 0.70 | **0.20** | 2.30 | 2.81 | **0.28** | 1.62 | 2.61 | **0.23** | 1.51 | 1.53 | **0.23** | 1.15 | 1.62 |
| 4 | **0.06** | 0.18 | 0.62 | **0.08** | 0.19 | 0.67 | **0.10** | 3.88 | 1.79 | **0.12** | 2.00 | 1.64 | **0.18** | 0.41 | 3.31 | **0.20** | 0.46 | 2.06 |
| 5 | **0.05** | 0.04 | 0.08 | **0.06** | **0.06** | 0.15 | **0.11** | 1.00 | 2.56 | **0.15** | 1.31 | 2.56 | **0.08** | 0.14 | 0.42 | **0.08** | 0.14 | 0.38 |
| 6 | **0.04** | **0.04** | 0.06 | **0.04** | 0.05 | 0.06 | **0.10** | 0.41 | 0.70 | **0.11** | 0.47 | 0.54 | **0.06** | 0.12 | 0.24 | **0.07** | 0.10 | 0.25 |
| 7 | 0.06 | **0.05** | 0.07 | **0.06** | **0.06** | 0.08 | **0.10** | 0.86 | 1.56 | **0.09** | 1.08 | 0.80 | **0.07** | 0.12 | 0.26 | **0.06** | 0.09 | 0.15 |
| 8 | **0.03** | 0.04 | 0.05 | **0.04** | **0.04** | **0.04** | **0.09** | 0.23 | 0.19 | **0.10** | 0.28 | 0.18 | **0.06** | 0.12 | 0.11 | **0.06** | 0.10 | 0.09 |

| #Ins | LLaMA-Adapter-v2 CHAIR$_S$ | | | CHAIR$_I$ | | | Lynx CHAIR$_S$ | | | CHAIR$_I$ | | | InternLM-XC CHAIR$_S$ | | | CHAIR$_I$ | | |
|---|---|---|---|---|---|---|---|---|---|---|---|---|---|---|---|---|---|---|
| | L$_1$ | L$_2$ | L$_3$ | L$_1$ | L$_2$ | L$_3$ | L$_1$ | L$_2$ | L$_3$ | L$_1$ | L$_2$ | L$_3$ | L$_1$ | L$_2$ | L$_3$ | L$_1$ | L$_2$ | L$_3$ |
| 3 | **0.80** | 2.23 | 1.76 | **0.75** | 1.97 | 1.69 | **0.14** | 0.35 | 0.36 | **0.19** | 0.40 | 0.40 | **0.13** | 1.13 | 1.91 | **0.13** | 2.35 | 5.83 |
| 4 | **0.11** | 0.75 | 1.03 | **0.13** | 1.02 | 1.06 | **0.06** | 0.12 | 12.90 | **0.07** | 0.15 | 0.84 | **0.06** | 0.20 | 1.78 | **0.07** | 0.20 | 1.13 |
| 5 | **0.09** | 0.33 | 0.60 | **0.10** | 0.39 | 0.79 | **0.05** | 0.14 | 0.22 | **0.07** | 0.15 | 0.14 | **0.08** | 0.23 | 0.67 | **0.08** | 1.08 | 0.54 |
| 6 | **0.09** | 0.19 | 0.43 | **0.10** | 0.27 | 0.51 | **0.03** | 0.04 | 0.08 | **0.05** | 0.07 | 0.13 | **0.06** | 0.12 | 0.14 | **0.05** | 0.10 | 0.21 |
| 7 | **0.07** | 0.09 | 0.10 | **0.08** | 0.12 | 0.15 | **0.02** | 0.05 | 0.13 | **0.04** | 0.08 | 0.14 | **0.05** | 0.12 | 0.12 | **0.05** | 0.09 | 0.08 |
| 8 | **0.07** | 0.10 | 0.20 | **0.08** | 0.14 | 0.27 | **0.03** | 0.06 | 0.05 | **0.05** | 0.10 | 0.08 | **0.05** | 0.10 | 0.12 | **0.05** | 0.10 | 0.10 |

| #Ins | mPLUG-Owl CHAIR$_S$ | | | CHAIR$_I$ | | | Otter CHAIR$_S$ | | | CHAIR$_I$ | | | VPGTrans CHAIR$_S$ | | | CHAIR$_I$ | | |
|---|---|---|---|---|---|---|---|---|---|---|---|---|---|---|---|---|---|---|
| | L$_1$ | L$_2$ | L$_3$ | L$_1$ | L$_2$ | L$_3$ | L$_1$ | L$_2$ | L$_3$ | L$_1$ | L$_2$ | L$_3$ | L$_1$ | L$_2$ | L$_3$ | L$_1$ | L$_2$ | L$_3$ |
| 3 | **0.06** | 0.93 | 1.22 | **0.08** | 0.67 | 0.80 | **0.14** | 0.49 | 0.45 | **0.13** | 0.59 | 0.49 | **0.14** | 3.27 | 1.84 | **0.17** | 16.25 | 5.50 |
| 4 | **0.03** | 0.12 | 1.12 | **0.07** | 0.10 | 2.80 | **0.07** | 0.13 | 0.83 | **0.07** | 0.12 | 0.35 | **0.13** | 0.87 | 2.66 | **0.14** | 0.85 | 5.69 |
| 5 | **0.02** | 0.10 | 0.53 | **0.06** | 0.08 | 0.77 | **0.09** | 0.12 | 0.17 | **0.08** | 0.12 | 0.16 | **0.09** | 0.24 | 1.06 | **0.10** | 0.16 | 0.96 |
| 6 | **0.02** | 0.05 | 0.05 | **0.04** | 0.07 | 0.08 | **0.09** | 0.10 | **0.09** | 0.09 | 0.11 | **0.08** | **0.07** | 0.14 | 0.45 | **0.05** | 0.10 | 0.25 |
| 7 | **0.02** | 0.04 | 0.07 | **0.04** | 0.06 | 0.08 | **0.05** | 0.08 | 0.10 | **0.05** | 0.09 | 0.09 | **0.05** | 0.13 | 0.22 | 0.06 | **0.05** | 0.10 |
| 8 | **0.02** | 0.05 | 0.06 | **0.04** | 0.06 | 0.05 | **0.06** | 0.09 | 0.10 | **0.06** | 0.09 | 0.09 | **0.04** | 0.11 | 0.14 | **0.05** | 0.06 | 0.07 |

| #Ins | Qwen-VL CHAIR$_S$ | | | CHAIR$_I$ | | | mPLUG-Owl2 CHAIR$_S$ | | | CHAIR$_I$ | | | Gemini-Pro-V CHAIR$_S$ | | | CHAIR$_I$ | | |
|---|---|---|---|---|---|---|---|---|---|---|---|---|---|---|---|---|---|---|
| | L$_1$ | L$_2$ | L$_3$ | L$_1$ | L$_2$ | L$_3$ | L$_1$ | L$_2$ | L$_3$ | L$_1$ | L$_2$ | L$_3$ | L$_1$ | L$_2$ | L$_3$ | L$_1$ | L$_2$ | L$_3$ |
| 3 | **0.09** | 1.11 | 0.99 | **0.11** | 2.26 | 1.67 | **0.27** | 0.88 | 0.81 | **0.34** | 0.91 | 0.91 | **0.18** | 1.02 | 1.03 | **0.25** | 3.92 | 1.24 |
| 4 | **0.15** | 0.40 | 4.94 | **0.11** | 0.43 | 19.84 | **0.09** | 0.67 | 4.33 | **0.05** | 0.39 | 1.76 | **0.13** | 0.19 | 1.67 | **0.16** | 0.24 | 1.87 |
| 5 | **0.05** | 0.05 | 3.40 | **0.08** | 0.14 | 2.23 | **0.06** | 0.20 | 4.27 | **0.04** | 0.24 | 0.84 | **0.12** | 2.28 | 0.51 | **0.15** | 1.27 | 1.18 |
| 6 | **0.06** | 0.09 | 0.74 | **0.07** | 0.12 | 0.44 | **0.04** | 0.08 | 0.33 | **0.03** | 0.10 | 0.85 | **0.10** | 1.00 | 0.84 | **0.09** | 0.38 | 0.47 |
| 7 | **0.05** | 0.08 | 0.93 | **0.06** | 0.08 | 0.89 | **0.05** | 0.09 | 0.30 | **0.03** | 0.15 | 0.23 | **0.11** | 0.09 | 0.15 | **0.12** | 0.13 | 0.30 |
| 8 | **0.04** | 0.06 | 0.16 | **0.06** | 0.08 | 0.19 | **0.04** | 0.08 | 33.42 | **0.04** | 0.11 | 0.20 | **0.08** | 0.08 | 0.13 | **0.09** | 0.11 | 0.12 |

Table 3: The average RSD of CHAIR scores with the LeHaCE on MSCOCO with different fitting methods. L$_1$ represents LeHaCE with linear fitting, while L$_2$ and L$_3$ represent LeHaCE with quadratic and cubic polynomial fitting, respectively.

the results. For the number of instructions in each instruction set, we conduct extensive experiments under five different conditions: 3, 4, 5, 6, 7, and 8. The experiments are carried out 10 times using distinct instruction sets, and the final results are determined by averaging the outcomes of these 10 experiments.

The results are shown in Table 2, from which we can observe that LeHaCE demonstrates superior stability compared to the average-based framework in nearly all cases. Notably, **1)** On the MSCOCO dataset, for the CHAIR$_I$ metric, LeHaCE consistently outperforms the average-based framework across all twelve LVLMs when the number of instructions reaches five or more. Similarly, for the CHAIR$_S$ metric, LeHaCE exhibits superior performance across all twelve LVLMs when the number of instructions reaches four or more. **2)** On the NoCaps dataset, when the number of instructions reaches four or more, LeHaCE consistently outperforms the average-based framework across all twelve LVLMs on both CHAIR$_I$ and CHAIR$_S$ metrics. In Table 2, we observe that when the number of instructions is very low, such as three, the stability of LeHaCE is compromised due to the difficulty in accurately fitting the length-hallucination curve. However, with just four or five instructions, LeHaCE consistently exhibits superior stability.

To verify the validity of the linear assumption we make in the LeHaCE method, we evaluate the stability of LeHaCE with different fitting methods. Specifically, we assess the RSD of LeHaCE on the MSCOCO dataset when applying linear, quadratic, and cubic fitting. As shown in Table 3, the results indicate that linear fitting significantly outperforms polynomial fitting, particularly when the instruction count is low.

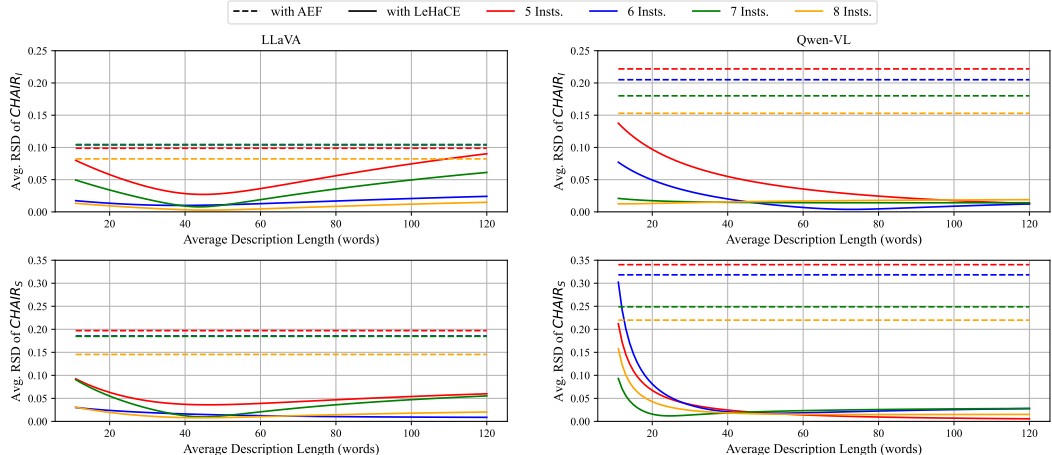

Figure 5: Average RSD of CHAIR with the LeHaCE framework at different lengths, lower is better. ABF refers to the average-based evaluation framework.

For the stability of LeHaCE at different lengths, the results are shown in Figure 5, from which we can see that LeHaCE significantly improves the stability of the CHAIR metrics across a wide range of description lengths. All of these experimental results validate the superior stability of LeHaCE.

## 5 Conclusion and Limitations

**Conclusion**: In this paper, we find the degree of object hallucinations is primarily influenced by the length of image descriptions, with instructions only indirectly affecting hallucinations through their effect on image description lengths. The degree of object hallucination and the length of image descriptions exhibit a clear positive linear correlation. Based on our findings, a stable, fair and comprehensive object hallucination evaluation framework named LeHaCE is introduced. Extensive experimental results validate the superiority of LeHaCE over existing frameworks.

**Limitations**: Despite exhaustive investigations, this work still has potential limitations. **1)** We focus on object hallucination, leaving other types of hallucinations for future work. **2)** Due to computational constraints, we evaluate LVLMs on only a subset of each dataset. Nevertheless, we conduct thorough experiments across various datasets to validate our findings and method. **3)** Due to high API fees, we only explore one proprietary business LVLM in our experiments. However, we conduct in-depth analyses on eleven open-source LVLMs, validating the broad applicability of our method. **4)** In the typical practice of evaluating hallucination levels in LVLMs, multiple instructions are usually used to enhance the stability of the evaluation results. Although LeHaCE cannot be used with just one instruction, this limitation does not affect its ability to provide stable evaluations.

## 6 Acknowledgments

This work was supported in part by the National Key R&D Program of China (2021YFF0900500), and the National Natural Science Foundation of China (NSFC) under grants U22B2035 and 62441202.

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

# 7 Technical Appendices

## 7.1 Instructions

We primarily referred to the instructions from [4] and additionally designed some instructions, totaling 25 in number.

- $I_1$:'Describe the image in one sentence.',
- $I_2$:'Summarize the image in a single sentence.',
- $I_3$:'Give a one-sentence depiction of the image.',
- $I_4$:'Provide a concise sentence describing the image.',
- $I_5$:'Give a brief summary of the image in a single sentence.',
- $I_6$:'Describe this image in short.',
- $I_7$:'Describe this image in a few words.',
- $I_8$:'Provide a brief caption for this image.',
- $I_9$:'Provide a short caption for this image.',
- $I_{10}$:'Briefly describe the content of the image.',
- $I_{11}$:'Describe this image.',
- $I_{12}$:'What does the image show?',
- $I_{13}$:'What can you see in the image?',
- $I_{14}$:'What is described in the image?',
- $I_{15}$:'Provide a caption for this image.',
- $I_{16}$:'Describe the objects in this image.',
- $I_{17}$:'Can you provide a description of the image?',
- $I_{18}$:'What objects or subjects are present in the image?',
- $I_{19}$:'Describe this image in detail.',
- $I_{20}$:'Describe this image in extremely detail.',
- $I_{21}$:'Provide a detailed description of this image.',
- $I_{22}$:'Can you describe the scene in the image in great detail?',
- $I_{23}$:'Give a thorough account of what is depicted in this image.',
- $I_{24}$:'Provide an elaborate and comprehensive analysis of this image.',
- $I_{25}$:'Give a comprehensive and in-depth description of what is shown in this image.',

## 7.2 Further Exploration

Why does the hallucination rate of MLLMs increase with the increase in image description length? The underlying reasons behind this phenomenon are difficult to determine, as the output of MLLMs is influenced by multiple factors such as visual encoders, language models, and training data. In this section, we aim to shed light on this phenomenon by delving into an analysis of common hallucination patterns in long image descriptions.

As shown in Figure 7, We found that hallucinations are more likely to occur after some words or phrases that indicate enumeration or introduce additional information, such as "in addition", "addition to", "additionally", "include", "including", "such as", "as well" and "also". We refer to these words/phrases as "hallucinogenic words". As shown in Figure 6 left, we performed a statistical analysis on a subset of MSCOCO dataset comprising 256 images to investigate the hallucination rate of image descriptions containing hallucinogenic words. The experimental results demonstrate a notable increase in the hallucination rate of image descriptions that incorporate hallucinogenic words, compared to descriptions that lack such words. This phenomenon was consistently observed across all twelve MLLMs.

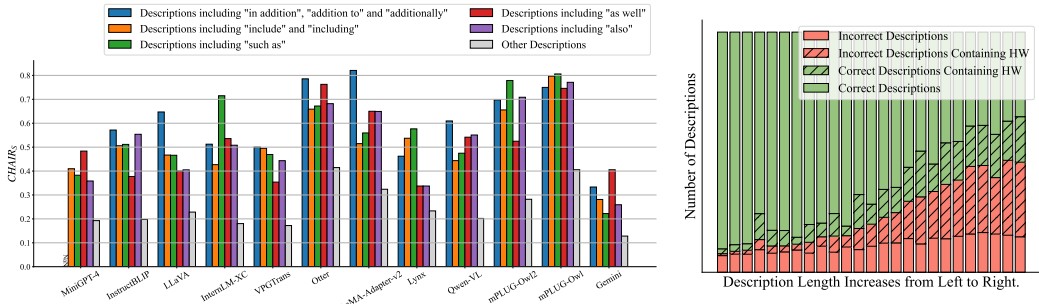

Figure 6: **Left**: Comparison of the hallucination rates between image descriptions containing hallucinogenic words and image descriptions without hallucinogenic words. **Right**: Proportion of hallucinogenic words in image descriptions containing hallucinations under different instructions.

Therefore, we propose a hypothesis that MLLMs are more likely to employ hallucinogenic words in generating lengthy and detailed image descriptions, resulting in a higher hallucination rate. To validate our hypothesis, we explored the relationship between the occurrence frequency of hallucinogenic words and the average length of image descriptions. The image descriptions with varying average lengths were generated by different instructions. The results are depicted in Figure 6 right. As the length of the description increases, the number of incorrect descriptions containing hallucinogenic words increases and constitutes a significant portion of all incorrect descriptions. This validates our hypothesis."

| Metric | MSCOCO | | NoCaps | |
|---|---|---|---|---|
| | Beam Search | Beam Search w/o HW | Beam Search | Beam Search w/o HW |
| $C_I \downarrow$ | 8.86 | **7.86** | 15.44 | **14.73** |
| $L_{C_I}(20) \downarrow$ | 5.15 | **4.77** | 11.43 | **11.14** |
| $L_{C_I}(40) \downarrow$ | 7.95 | **6.85** | 14.24 | **13.72** |
| $L_{C_I}(60) \downarrow$ | 10.04 | **8.93** | 17.05 | **16.31** |
| $L_{C_I}(80) \downarrow$ | 12.49 | **11.01** | 19.87 | **18.89** |
| $L_{C_I GR} \downarrow$ | 0.12 | **0.10** | 0.14 | **0.13** |
| $C_S \downarrow$ | 24.27 | **21.77** | 37.91 | **36.39** |
| $L_{C_S}(20) \downarrow$ | 9.69 | **8.93** | 21.15 | **20.49** |
| $L_{C_S}(40) \downarrow$ | 19.38 | **17.76** | 33.00 | **32.12** |
| $L_{C_S}(60) \downarrow$ | 29.07 | **26.60** | 44.85 | **43.75** |
| $L_{C_S}(80) \downarrow$ | 38.75 | **35.43** | 56.70 | **55.37** |
| $L_{C_S GR} \downarrow$ | 0.48 | **0.44** | 0.59 | **0.58** |
| T2I-CLIPRetrieval (R@1) ↑ | **32.75** | 32.73 | 46.39 | **48.49** |
| CLIPScore ↑ | 0.81 | 0.81 | 0.80 | 0.80 |
| RefCLIPScore ↑ | 0.81 | 0.81 | 0.81 | **0.82** |

Table 4: Hallucination rate and quality of the generated image descriptions after disabling hallucinogenic words. The values in the table are averaged across 10 MLLMs.

We further explored whether the hallucination rate of MLLMs can be reduced by disabling hallucinogenic words. Specifically, we prohibited the use of previously mentioned hallucinogenic words ("in addition," "addition to," "additionally," "include," "includes," "including," "such as," "as well," "also") during the generation process in the MLLMs, ensuring that the generated image descriptions did not contain these hallucinogenic words. The results, presented in Table 4, clearly demonstrate that disabling hallucinogenic words can significantly decrease the hallucination rate without compromising the quality of image descriptions. Furthermore, it helps alleviate the tendency for the hallucination rate to increase as the description length grows. These experimental findings not only highlight the substantial impact of hallucinogenic words on the hallucination rate of MLLMs but also offer valuable insights for mitigating hallucination in MLLMs.

### 7.3 More Experimental Results

The average length and CHAIR scores of image descriptions generated by 12 MLLMs (Gemini-Pro-Vision pro [2], Qwen-VL [7], MiniGPT-4 [5], LLaVA [6], InstructBLIP [4], LLaMA-Adapter-v2 [41], mPLUG-Owl2 [36], mPLUG-Owl [38], InternLM-XComposer [32], VPGTrans [42], Otter [43] and Lynx [44]) under 25 instructions on subsets of MSCOCO [45] and NoCap [46] are shown in Table 5 and Table 6, respectively.

The hallucination degree of each MLLM with hallucinogenic words disabled are shown in Table 4 and Table 8. Disabling hallucinogenic terms can effectively alleviate hallucinations in MLLMs.

### 7.4 More Case Studies

Figure 7 showcases examples of image descriptions generated by MLLMs before and after disabling hallucinogenic words.

Table 5 (top half):

| Ins | MiniGPT-4 | | | InstructBLIP | | | Lynx | | | LLaVA | | | Otter | | | VPGTrans | | |
|---|---|---|---|---|---|---|---|---|---|---|---|---|---|---|---|---|---|---|
| | Len | $C_I$ | $C_S$ | Len | $C_I$ | $C_S$ | Len | $C_I$ | $C_S$ | Len | $C_I$ | $C_S$ | Len | $C_I$ | $C_S$ | Len | $C_I$ | $C_S$ |
| $I_1$ | 17.29 | 3.76 | 6.64 | 43.52 | 5.63 | 14.84 | 34.97 | 6.86 | 17.58 | 18.70 | 7.40 | 14.84 | 26.48 | 10.78 | 20.31 | 15.92 | 3.75 | 6.25 |
| $I_2$ | 17.62 | 2.99 | 5.08 | 11.39 | 1.18 | 1.56 | 38.17 | 4.70 | 12.50 | 19.41 | 8.02 | 16.02 | 32.30 | 12.35 | 26.17 | 15.20 | 5.17 | 7.81 |
| $I_3$ | 20.83 | 5.54 | 9.38 | 10.96 | 0.97 | 1.56 | 36.69 | 6.10 | 15.23 | 18.12 | 5.33 | 10.16 | 27.01 | 10.95 | 20.70 | 15.30 | 4.45 | 6.25 |
| $I_4$ | 25.26 | 5.06 | 8.59 | 11.00 | 0.96 | 1.56 | 35.68 | 4.91 | 13.28 | 18.69 | 6.79 | 14.06 | 51.97 | 13.33 | 35.94 | 16.55 | 5.02 | 7.42 |
| $I_5$ | 21.76 | 5.31 | 9.77 | 10.82 | 0.95 | 1.56 | 38.19 | 6.50 | 18.36 | 21.14 | 6.91 | 14.06 | 35.20 | 12.04 | 25.39 | 15.65 | 4.94 | 7.42 |
| $I_6$ | 51.10 | 8.70 | 22.27 | 84.66 | 9.50 | 33.59 | 38.46 | 7.37 | 19.92 | 46.25 | 9.87 | 26.95 | 69.41 | 18.35 | 48.83 | 49.00 | 8.06 | 20.31 |
| $I_7$ | 39.39 | 6.36 | 16.02 | 76.02 | 7.81 | 28.52 | 40.21 | 6.75 | 17.19 | 31.38 | 6.98 | 16.41 | 58.12 | 14.33 | 40.63 | 24.34 | 7.31 | 12.89 |
| $I_8$ | 48.46 | 7.75 | 19.53 | 11.01 | 0.96 | 1.56 | 37.14 | 6.36 | 17.19 | 45.61 | 9.00 | 21.09 | 54.59 | 13.33 | 33.98 | 23.71 | 5.97 | 8.59 |
| $I_9$ | 42.79 | 6.85 | 16.80 | 10.86 | 1.20 | 1.95 | 36.79 | 5.56 | 14.84 | 45.55 | 9.08 | 21.48 | 53.70 | 13.48 | 33.59 | 16.83 | 4.20 | 5.47 |
| $I_{10}$ | 50.16 | 7.95 | 21.88 | 10.92 | 1.21 | 1.95 | 44.59 | 7.10 | 19.14 | 27.44 | 6.92 | 14.06 | 57.01 | 14.34 | 41.80 | 46.24 | 7.78 | 17.97 |
| $I_{11}$ | 63.67 | 7.67 | 23.44 | 90.46 | 10.02 | 35.55 | 46.65 | 6.85 | 19.92 | 57.75 | 8.96 | 27.34 | 78.25 | 21.68 | 62.11 | 68.75 | 8.19 | 22.27 |
| $I_{12}$ | 34.10 | 6.74 | 14.06 | 24.25 | 4.10 | 8.98 | 70.71 | 10.15 | 29.69 | 19.06 | 6.88 | 13.28 | 1.63 | 1.57 | 1.17 | 23.51 | 6.58 | 12.50 |
| $I_{13}$ | 41.19 | 6.93 | 16.41 | 40.76 | 4.94 | 13.28 | 68.39 | 11.36 | 31.25 | 26.33 | 7.99 | 16.41 | 51.03 | 16.36 | 40.23 | 39.59 | 7.91 | 17.97 |
| $I_{14}$ | 41.82 | 7.75 | 19.53 | 26.76 | 4.24 | 9.38 | 66.39 | 10.57 | 28.13 | 20.92 | 5.60 | 10.94 | 16.19 | 8.53 | 10.94 | 26.46 | 7.18 | 12.50 |
| $I_{15}$ | 40.84 | 7.84 | 17.97 | 11.11 | 0.95 | 1.56 | 41.04 | 6.67 | 17.58 | 60.71 | 7.42 | 18.75 | 63.95 | 15.54 | 35.94 | 19.73 | 5.18 | 7.03 |
| $I_{16}$ | 55.95 | 9.05 | 21.48 | 81.48 | 8.72 | 34.77 | 50.32 | 8.10 | 23.05 | 33.21 | 10.80 | 23.05 | 60.63 | 20.29 | 55.47 | 58.06 | 9.44 | 24.61 |
| $I_{17}$ | 67.46 | 7.79 | 22.66 | 11.02 | 0.96 | 1.56 | 58.48 | 9.82 | 26.95 | 41.94 | 8.23 | 20.70 | 59.64 | 14.71 | 39.06 | 46.72 | 8.25 | 10.55 |
| $I_{18}$ | 29.29 | 6.62 | 11.33 | 18.74 | 3.12 | 5.86 | 61.49 | 13.40 | 32.81 | 18.98 | 8.58 | 19.14 | 23.50 | 15.92 | 32.81 | 21.18 | 6.46 | 10.94 |
| $I_{19}$ | 75.31 | 7.31 | 24.22 | 98.68 | 14.29 | 52.34 | 60.34 | 8.68 | 26.95 | 84.97 | 10.48 | 35.55 | 81.89 | 20.26 | 61.33 | 93.45 | 8.83 | 28.52 |
| $I_{20}$ | 87.63 | 8.42 | 28.13 | 100.46 | 16.05 | 55.86 | 61.69 | 9.23 | 27.73 | 94.15 | 11.56 | 37.11 | 86.79 | 21.36 | 63.28 | 102.59 | 10.16 | 33.59 |
| $I_{21}$ | 77.58 | 9.04 | 27.34 | 94.07 | 14.57 | 51.56 | 53.25 | 9.06 | 25.78 | 89.18 | 11.34 | 36.33 | 82.18 | 21.09 | 66.80 | 92.48 | 9.80 | 29.69 |
| $I_{22}$ | 80.14 | 11.14 | 32.42 | 100.25 | 16.70 | 58.20 | 84.23 | 14.88 | 39.45 | 81.75 | 12.71 | 37.89 | 70.52 | 13.41 | 41.02 | 85.76 | 10.29 | 28.13 |
| $I_{23}$ | 74.70 | 9.87 | 26.95 | 12.10 | 1.34 | 1.95 | 72.46 | 11.17 | 32.03 | 60.71 | 7.42 | 18.75 | 63.95 | 21.49 | 58.20 | 79.47 | 9.05 | 29.30 |
| $I_{24}$ | 94.29 | 8.51 | 29.69 | 96.74 | 11.44 | 43.36 | 78.71 | 11.99 | 35.55 | 112.24 | 12.01 | 37.89 | 90.93 | 26.89 | 80.47 | 101.60 | 8.54 | 31.25 |
| $I_{25}$ | 78.88 | 9.85 | 30.47 | 77.02 | 8.46 | 30.86 | 53.86 | 8.02 | 21.48 | 96.59 | 10.62 | 37.50 | 83.79 | 21.50 | 60.16 | 85.68 | 9.40 | 30.47 |

| Ins | LLaMA-Adapter-v2 | | | mPLUG-Owl | | | Gemini-Pro-Vision | | | InternLM-XComposer | | | Qwen-VL | | | mPLUG-Owl2 | | |
|---|---|---|---|---|---|---|---|---|---|---|---|---|---|---|---|---|---|---|
| | Len | $C_I$ | $C_S$ | Len | $C_I$ | $C_S$ | Len | $C_I$ | $C_S$ | Len | $C_I$ | $C_S$ | Len | $C_I$ | $C_S$ | Len | $C_I$ | $C_S$ |
| $I_1$ | 30.59 | 8.56 | 18.36 | 22.71 | 7.19 | 12.50 | 15.13 | 4.00 | 6.25 | 14.17 | 4.23 | 7.03 | 18.10 | 2.26 | 4.69 | 9.96 | 2.37 | 2.73 |
| $I_2$ | 23.19 | 7.02 | 15.23 | 18.71 | 5.26 | 9.77 | 14.06 | 3.90 | 5.86 | 13.44 | 3.90 | 6.25 | 18.04 | 3.08 | 5.86 | 10.04 | 2.33 | 3.13 |
| $I_3$ | 20.00 | 7.22 | 13.67 | 15.98 | 4.36 | 7.42 | 14.70 | 3.84 | 6.25 | 13.98 | 3.93 | 6.64 | 18.89 | 3.00 | 5.86 | 10.04 | 2.37 | 3.13 |
| $I_4$ | 18.61 | 6.70 | 12.89 | 15.36 | 3.89 | 6.25 | 14.54 | 3.19 | 5.86 | 14.93 | 4.46 | 7.42 | 18.34 | 3.02 | 5.86 | 9.93 | 2.35 | 3.13 |
| $I_5$ | 32.91 | 9.76 | 21.09 | 23.90 | 6.81 | 12.89 | 15.41 | 2.62 | 4.30 | 13.90 | 4.31 | 7.42 | 18.00 | 2.55 | 4.69 | 10.00 | 2.10 | 2.73 |
| $I_6$ | 79.85 | 19.07 | 52.73 | 87.25 | 21.27 | 57.42 | 19.84 | 2.92 | 5.47 | 22.51 | 7.07 | 11.33 | 49.76 | 6.03 | 17.58 | 64.30 | 12.32 | 33.98 |
| $I_7$ | 72.24 | 16.46 | 44.92 | 72.70 | 18.32 | 48.83 | 10.77 | 1.70 | 2.34 | 22.23 | 4.77 | 9.38 | 27.00 | 4.22 | 9.77 | 41.52 | 8.96 | 21.09 |
| $I_8$ | 60.97 | 11.34 | 31.25 | 32.95 | 12.30 | 20.70 | 14.33 | 4.65 | 6.64 | 19.11 | 6.57 | 9.77 | 25.70 | 4.98 | 9.77 | 9.73 | 2.43 | 3.13 |
| $I_9$ | 46.49 | 10.43 | 26.17 | 16.91 | 6.21 | 9.38 | 12.78 | 3.14 | 3.52 | 17.54 | 4.83 | 7.03 | 19.36 | 5.63 | 9.38 | 9.72 | 2.39 | 3.13 |
| $I_{10}$ | 44.54 | 8.68 | 22.27 | 74.50 | 17.38 | 45.31 | 26.22 | 3.50 | 8.20 | 19.41 | 4.17 | 6.64 | 49.01 | 5.89 | 19.14 | 70.53 | 13.27 | 39.06 |
| $I_{11}$ | 87.73 | 19.38 | 58.98 | 101.23 | 21.65 | 59.77 | 42.68 | 6.18 | 15.23 | 59.15 | 9.73 | 25.78 | 75.04 | 8.47 | 28.91 | 91.07 | 17.04 | 57.03 |
| $I_{12}$ | 20.29 | 5.01 | 8.98 | 82.36 | 18.70 | 47.66 | 28.59 | 6.07 | 12.11 | 40.42 | 6.94 | 17.58 | 19.64 | 2.50 | 4.69 | 16.80 | 3.03 | 5.08 |
| $I_{13}$ | 29.50 | 4.96 | 11.33 | 70.57 | 17.19 | 41.41 | 33.11 | 5.48 | 14.06 | 38.90 | 8.28 | 18.36 | 28.87 | 3.74 | 6.64 | 30.76 | 4.45 | 10.94 |
| $I_{14}$ | 18.46 | 4.75 | 8.98 | 88.11 | 18.30 | 52.34 | 31.86 | 5.46 | 12.11 | 27.32 | 6.46 | 11.72 | 42.22 | 5.58 | 15.63 | 34.97 | 5.77 | 13.67 |
| $I_{15}$ | 69.86 | 11.92 | 34.77 | 46.09 | 14.23 | 26.17 | 18.38 | 4.43 | 5.08 | 17.88 | 5.94 | 9.77 | 48.83 | 7.73 | 20.31 | 9.80 | 2.40 | 3.13 |
| $I_{16}$ | 53.31 | 12.15 | 31.25 | 64.19 | 18.58 | 40.63 | 45.50 | 6.45 | 16.02 | 56.87 | 10.90 | 32.03 | 66.16 | 7.47 | 26.95 | 70.91 | 12.04 | 39.06 |
| $I_{17}$ | 59.07 | 8.18 | 25.78 | 96.80 | 21.08 | 61.33 | 50.84 | 7.69 | 18.36 | 37.50 | 8.99 | 18.36 | 66.72 | 8.30 | 26.95 | 83.73 | 14.85 | 50.39 |
| $I_{18}$ | 19.68 | 9.59 | 20.70 | 19.64 | 8.86 | 13.67 | 30.73 | 6.83 | 12.89 | 31.91 | 8.78 | 20.31 | 17.07 | 3.00 | 6.25 | 16.44 | 2.82 | 5.08 |
| $I_{19}$ | 89.76 | 19.98 | 60.55 | 101.93 | 22.28 | 64.84 | 62.08 | 6.05 | 19.92 | 66.83 | 10.52 | 32.81 | 87.10 | 9.58 | 36.72 | 93.40 | 16.99 | 63.67 |
| $I_{20}$ | 90.54 | 20.21 | 59.77 | 103.02 | 21.63 | 64.45 | 85.19 | 7.93 | 25.39 | 71.45 | 11.06 | 32.03 | 92.94 | 9.78 | 39.45 | 96.84 | 18.89 | 67.19 |
| $I_{21}$ | 88.58 | 19.37 | 58.98 | 100.25 | 21.52 | 61.72 | 63.95 | 6.35 | 18.75 | 72.29 | 11.64 | 35.55 | 86.27 | 10.87 | 39.06 | 90.73 | 15.76 | 59.77 |
| $I_{22}$ | 84.65 | 8.48 | 31.25 | 105.22 | 22.39 | 60.55 | 79.56 | 9.56 | 31.25 | 49.08 | 7.29 | 19.53 | 87.15 | 7.75 | 28.13 | 80.87 | 13.73 | 44.92 |
| $I_{23}$ | 85.40 | 13.11 | 38.67 | 103.27 | 21.98 | 61.72 | 74.96 | 5.97 | 18.75 | 69.31 | 11.94 | 35.55 | 85.61 | 12.44 | 44.92 | 89.39 | 15.99 | 55.47 |
| $I_{24}$ | 92.66 | 22.14 | 60.94 | 106.13 | 21.34 | 66.41 | 134.82 | 7.18 | 25.39 | 60.96 | 9.95 | 27.34 | 105.59 | 10.84 | 45.70 | 99.26 | 16.47 | 60.55 |
| $I_{25}$ | 83.82 | 13.31 | 40.23 | 102.21 | 21.93 | 60.94 | 81.99 | 6.51 | 21.09 | 69.60 | 11.75 | 35.16 | 95.62 | 10.95 | 42.19 | 92.32 | 15.90 | 53.91 |

Table 5: The average length and CHAIR scores of descriptions generated by 12 MLLMs prompted by the 25 instructions on a subset of MSCOCO containing 256 images.

## 7.5 Broader Impacts

The LeHaCE framework provides a stable, fair, and comprehensive way to evaluate object hallucinations in large vision-language models. It helps in assessing the usability of large vision-language models, thereby helping to prevent safety incidents.

We acknowledge the potential ethical concerns associated with the use of the MSCOCO dataset, particularly regarding data privacy, copyright, and consent, as the images in this dataset were collected from Flickr without explicit user consent. However, it is important to note that our study's methodology and findings are independent of the dataset used. Our research focuses on evaluating the relationship between instruction length and hallucinations in Large Vision-Language Models (LVLMs), and does not rely on or alter the underlying data in the MSCOCO dataset. Nonetheless, we recognize the ethical implications of using such datasets and recommend future research to continue exploring these issues in greater depth.

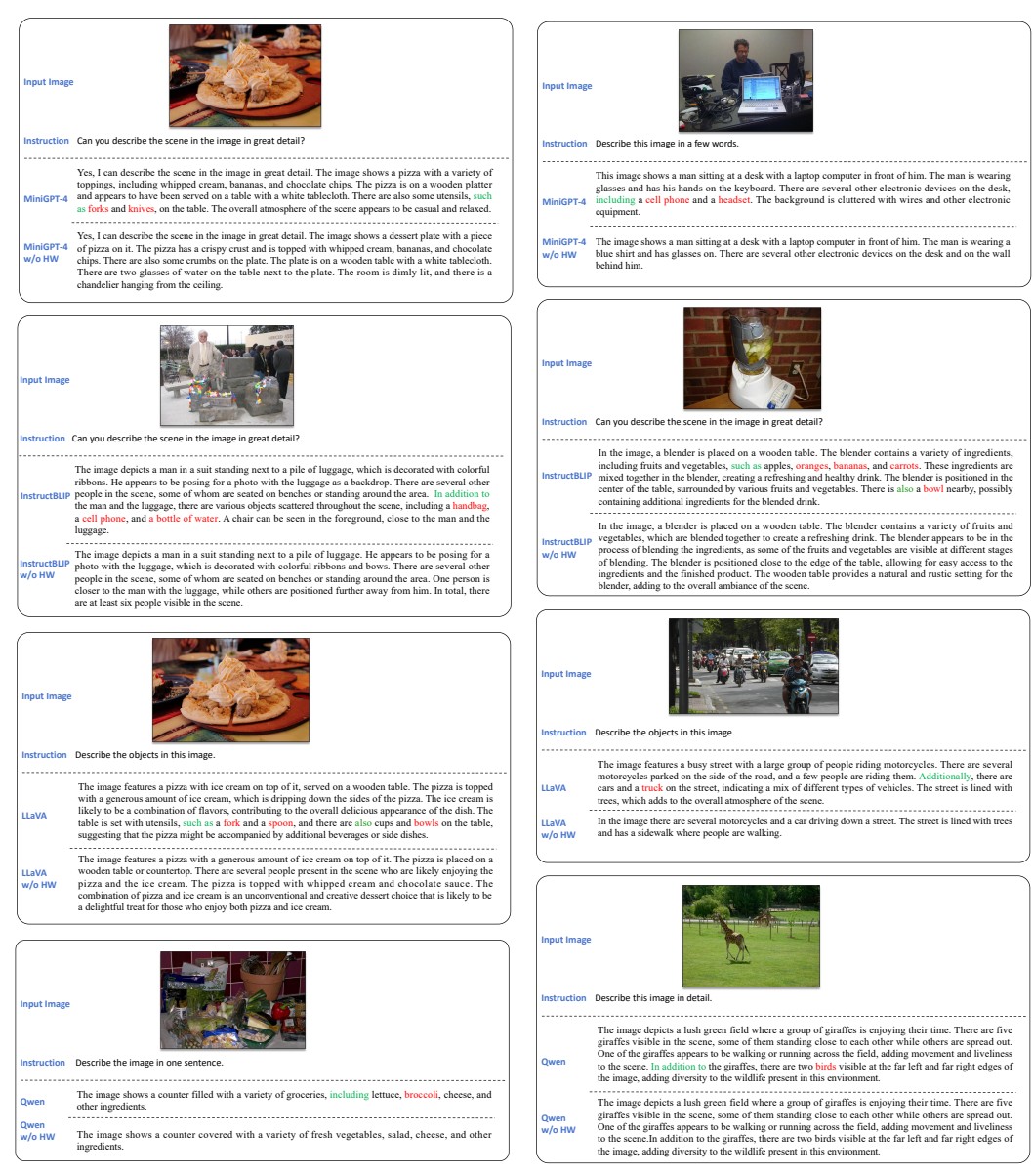

Figure 7: Example of detailed image descriptions generated by beam search and beam search without hallucinogenic words. The hallucination content is highlighted in red, and the hallucinogenic words are highlighted in green.

| Ins | MiniGPT-4 | | | InstructBLIP | | | Lynx | | | LLaVA | | | Otter | | | VPGTrans | | |
|---|---|---|---|---|---|---|---|---|---|---|---|---|---|---|---|---|---|---|
| | Len | $C_I$ | $C_S$ | Len | $C_I$ | $C_S$ | Len | $C_I$ | $C_S$ | Len | $C_I$ | $C_S$ | Len | $C_I$ | $C_S$ | Len | $C_I$ | $C_S$ |
| $I_1$ | 21.48 | 12.13 | 17.19 | 46.27 | 13.03 | 36.72 | 35.52 | 15.84 | 42.19 | 19.49 | 11.09 | 18.75 | 25.50 | 17.78 | 33.59 | 15.34 | 10.85 | 16.80 |
| $I_2$ | 24.50 | 13.58 | 21.88 | 11.50 | 4.29 | 5.86 | 39.05 | 17.65 | 48.05 | 20.04 | 12.23 | 23.05 | 31.52 | 19.63 | 35.55 | 15.16 | 8.85 | 12.89 |
| $I_3$ | 26.64 | 15.97 | 24.61 | 10.98 | 4.50 | 6.25 | 35.79 | 14.97 | 41.41 | 18.55 | 12.12 | 21.88 | 26.04 | 17.66 | 31.64 | 16.37 | 12.65 | 19.92 |
| $I_4$ | 34.26 | 16.03 | 29.69 | 11.09 | 4.51 | 6.25 | 35.30 | 16.16 | 42.19 | 18.84 | 11.03 | 19.53 | 49.61 | 21.83 | 46.09 | 17.63 | 10.98 | 17.19 |
| $I_5$ | 33.45 | 13.93 | 28.52 | 11.05 | 5.66 | 7.81 | 38.95 | 17.60 | 46.88 | 21.60 | 12.21 | 24.61 | 34.21 | 19.79 | 39.06 | 15.51 | 11.25 | 16.41 |
| $I_6$ | 59.70 | 19.80 | 53.13 | 80.65 | 17.69 | 55.08 | 40.11 | 16.78 | 47.66 | 45.34 | 14.64 | 35.94 | 68.98 | 24.04 | 56.64 | 47.39 | 17.76 | 45.70 |
| $I_7$ | 51.91 | 18.70 | 46.48 | 76.69 | 16.35 | 53.13 | 40.16 | 16.01 | 42.97 | 31.57 | 13.25 | 29.30 | 54.95 | 20.24 | 48.05 | 25.17 | 13.93 | 26.95 |
| $I_8$ | 60.38 | 18.72 | 51.56 | 10.98 | 4.23 | 5.86 | 35.79 | 16.47 | 43.36 | 46.28 | 17.59 | 40.63 | 53.29 | 19.40 | 40.23 | 24.84 | 11.75 | 19.14 |
| $I_9$ | 56.07 | 18.48 | 48.05 | 10.84 | 4.55 | 6.64 | 35.71 | 15.75 | 42.19 | 45.19 | 14.94 | 35.16 | 52.82 | 20.44 | 42.19 | 19.90 | 9.32 | 13.28 |
| $I_{10}$ | 60.73 | 21.23 | 52.73 | 10.81 | 4.85 | 6.64 | 43.50 | 17.53 | 48.83 | 28.38 | 15.55 | 31.25 | 55.16 | 19.78 | 44.53 | 46.40 | 17.65 | 47.66 |
| $I_{11}$ | 72.61 | 20.90 | 58.20 | 88.72 | 17.68 | 55.86 | 45.53 | 19.08 | 50.39 | 58.91 | 17.23 | 50.00 | 74.64 | 26.13 | 66.02 | 67.08 | 16.81 | 52.34 |
| $I_{12}$ | 47.92 | 18.85 | 43.36 | 23.92 | 8.53 | 17.97 | 69.55 | 23.03 | 60.94 | 19.81 | 11.52 | 20.31 | 3.64 | 7.46 | 5.86 | 23.54 | 13.41 | 23.44 |
| $I_{13}$ | 53.16 | 20.12 | 48.05 | 10.90 | 4.90 | 32.03 | 66.67 | 20.96 | 56.25 | 27.24 | 15.23 | 30.08 | 47.86 | 21.59 | 48.44 | 40.57 | 16.06 | 36.72 |
| $I_{14}$ | 52.89 | 18.69 | 46.09 | 25.81 | 9.16 | 18.75 | 62.31 | 18.86 | 53.13 | 21.47 | 10.57 | 21.09 | 17.83 | 13.02 | 17.97 | 27.65 | 14.37 | 27.73 |
| $I_{15}$ | 55.90 | 17.41 | 42.19 | 11.07 | 4.40 | 6.25 | 38.64 | 18.38 | 48.05 | 60.23 | 14.45 | 33.98 | 61.43 | 22.70 | 48.44 | 22.37 | 10.47 | 15.63 |
| $I_{16}$ | 64.24 | 21.43 | 55.47 | 77.42 | 16.38 | 51.17 | 45.67 | 19.44 | 50.78 | 33.66 | 16.15 | 37.50 | 60.77 | 24.13 | 58.20 | 58.08 | 18.43 | 49.22 |
| $I_{17}$ | 74.12 | 21.68 | 57.03 | 10.96 | 5.00 | 7.03 | 52.23 | 18.50 | 50.00 | 42.40 | 14.33 | 38.28 | 57.95 | 19.90 | 48.83 | 48.74 | 18.33 | 26.95 |
| $I_{18}$ | 41.50 | 18.43 | 39.84 | 17.88 | 6.10 | 10.94 | 63.18 | 23.39 | 62.11 | 20.17 | 14.15 | 27.73 | 23.75 | 23.24 | 43.36 | 21.86 | 15.38 | 22.66 |
| $I_{19}$ | 85.50 | 21.39 | 60.16 | 98.16 | 22.37 | 63.67 | 57.25 | 21.29 | 58.59 | 85.00 | 18.01 | 60.55 | 78.92 | 26.19 | 69.14 | 89.88 | 18.36 | 59.77 |
| $I_{20}$ | 92.09 | 21.91 | 62.11 | 97.32 | 23.13 | 66.02 | 58.61 | 19.74 | 53.91 | 93.33 | 18.49 | 63.28 | 82.53 | 26.52 | 66.02 | 99.56 | 19.29 | 64.06 |
| $I_{21}$ | 87.03 | 21.90 | 61.33 | 92.97 | 19.75 | 57.03 | 51.97 | 20.51 | 56.64 | 88.46 | 18.23 | 57.42 | 80.18 | 26.37 | 66.80 | 92.18 | 17.99 | 58.98 |
| $I_{22}$ | 84.68 | 23.02 | 64.45 | 97.89 | 20.72 | 60.55 | 76.86 | 23.08 | 62.50 | 83.05 | 20.32 | 63.28 | 67.57 | 19.85 | 49.22 | 83.74 | 19.68 | 54.30 |
| $I_{23}$ | 81.96 | 20.93 | 60.55 | 11.85 | 4.08 | 6.25 | 70.57 | 22.57 | 63.67 | 86.93 | 21.58 | 58.59 | 77.14 | 24.24 | 62.11 | 78.73 | 17.89 | 55.08 |
| $I_{24}$ | 98.97 | 20.35 | 61.72 | 95.41 | 19.07 | 60.16 | 76.41 | 22.29 | 64.45 | 110.76 | 20.01 | 59.38 | 89.67 | 30.58 | 77.34 | 98.40 | 16.61 | 59.38 |
| $I_{25}$ | 85.08 | 21.36 | 63.67 | 76.37 | 15.28 | 51.95 | 51.12 | 18.85 | 49.61 | 95.95 | 17.24 | 54.69 | 80.16 | 26.41 | 67.58 | 85.92 | 18.66 | 61.72 |

| Ins | LLaMA-Adapter-v2 | | | mPLUG-Owl | | | Gemini-Pro-Vision | | | InternLM-XComposer | | | Qwen-VL | | | mPLUG-Owl2 | | |
|---|---|---|---|---|---|---|---|---|---|---|---|---|---|---|---|---|---|---|
| | Len | $C_I$ | $C_S$ | Len | $C_I$ | $C_S$ | Len | $C_I$ | $C_S$ | Len | $C_I$ | $C_S$ | Len | $C_I$ | $C_S$ | Len | $C_I$ | $C_S$ |
| $I_1$ | 31.18 | 15.16 | 32.42 | 20.84 | 11.99 | 17.19 | 15.11 | 11.75 | 16.41 | 13.75 | 9.83 | 16.02 | 16.10 | 7.26 | 12.11 | 10.10 | 5.18 | 5.86 |
| $I_2$ | 22.96 | 14.04 | 25.00 | 17.77 | 12.18 | 18.36 | 14.24 | 8.36 | 11.33 | 13.16 | 8.99 | 15.63 | 16.28 | 8.35 | 12.50 | 10.00 | 5.52 | 6.25 |
| $I_3$ | 20.46 | 13.20 | 23.05 | 15.47 | 10.41 | 15.63 | 14.86 | 8.25 | 10.94 | 13.88 | 10.04 | 17.58 | 18.55 | 8.99 | 16.80 | 10.15 | 5.39 | 6.25 |
| $I_4$ | 19.14 | 12.11 | 20.70 | 15.17 | 10.50 | 13.67 | 14.49 | 10.39 | 14.45 | 14.41 | 9.50 | 16.02 | 18.11 | 8.13 | 14.06 | 9.85 | 5.54 | 6.25 |
| $I_5$ | 31.72 | 16.16 | 32.42 | 21.13 | 13.91 | 20.31 | 15.34 | 9.62 | 13.67 | 13.59 | 9.40 | 16.02 | 15.72 | 7.28 | 11.33 | 10.04 | 5.47 | 6.25 |
| $I_6$ | 79.44 | 25.74 | 62.89 | 79.79 | 21.98 | 56.64 | 18.66 | 13.11 | 19.14 | 18.95 | 12.76 | 21.88 | 25.48 | 10.61 | 17.58 | 32.86 | 12.77 | 22.27 |
| $I_7$ | 70.22 | 24.55 | 57.03 | 62.93 | 20.53 | 48.05 | 10.95 | 6.39 | 7.03 | 18.32 | 12.99 | 21.88 | 17.64 | 8.57 | 13.28 | 11.09 | 6.05 | 7.03 |
| $I_8$ | 58.44 | 20.65 | 49.22 | 29.20 | 16.16 | 26.95 | 14.89 | 8.82 | 11.33 | 16.70 | 12.09 | 19.92 | 19.23 | 5.95 | 8.98 | 9.57 | 4.79 | 5.08 |
| $I_9$ | 46.78 | 17.34 | 39.06 | 16.68 | 12.35 | 17.58 | 13.56 | 9.88 | 10.94 | 15.84 | 10.94 | 17.97 | 18.65 | 6.71 | 9.38 | 9.48 | 4.21 | 4.69 |
| $I_{10}$ | 43.18 | 16.77 | 36.33 | 61.12 | 18.17 | 42.97 | 24.44 | 14.17 | 28.13 | 17.54 | 11.35 | 20.70 | 29.62 | 13.06 | 24.61 | 51.38 | 14.11 | 31.64 |
| $I_{11}$ | 85.95 | 25.01 | 67.19 | 96.82 | 26.47 | 64.06 | 40.54 | 15.95 | 41.80 | 52.59 | 14.98 | 39.84 | 53.85 | 14.61 | 35.55 | 76.91 | 17.50 | 52.34 |
| $I_{12}$ | 20.60 | 11.07 | 18.36 | 66.96 | 20.69 | 45.70 | 28.32 | 14.89 | 27.34 | 30.93 | 11.06 | 24.61 | 17.44 | 6.31 | 9.38 | 13.86 | 4.71 | 5.47 |
| $I_{13}$ | 30.45 | 13.15 | 28.52 | 56.76 | 18.64 | 42.97 | 29.25 | 16.16 | 28.52 | 34.13 | 10.86 | 28.13 | 25.22 | 6.76 | 13.28 | 19.04 | 4.70 | 7.03 |
| $I_{14}$ | 18.62 | 10.85 | 16.80 | 73.85 | 19.16 | 50.00 | 29.42 | 15.77 | 30.47 | 19.32 | 7.88 | 12.50 | 42.02 | 11.67 | 26.95 | 18.46 | 5.40 | 7.42 |
| $I_{15}$ | 71.89 | 20.68 | 53.91 | 40.79 | 18.89 | 34.77 | 20.76 | 15.57 | 21.48 | 16.32 | 11.20 | 17.97 | 22.72 | 7.11 | 12.11 | 9.66 | 5.48 | 6.25 |
| $I_{16}$ | 56.18 | 18.08 | 49.61 | 49.82 | 22.37 | 42.97 | 40.91 | 18.18 | 47.66 | 52.25 | 15.86 | 41.02 | 59.89 | 13.72 | 39.06 | 58.22 | 14.74 | 39.84 |
| $I_{17}$ | 56.27 | 15.85 | 43.36 | 91.68 | 24.00 | 57.81 | 48.17 | 17.88 | 46.88 | 28.23 | 13.26 | 26.95 | 55.39 | 14.17 | 35.94 | 68.76 | 16.54 | 46.09 |
| $I_{18}$ | 19.98 | 14.67 | 28.91 | 16.64 | 9.15 | 12.50 | 30.06 | 18.27 | 39.84 | 29.03 | 11.79 | 25.39 | 18.59 | 8.42 | 14.45 | 14.90 | 3.59 | 5.08 |
| $I_{19}$ | 89.07 | 26.50 | 67.97 | 97.53 | 24.29 | 60.55 | 57.96 | 20.35 | 54.69 | 66.71 | 14.91 | 46.09 | 75.82 | 16.22 | 48.44 | 85.39 | 20.61 | 58.59 |
| $I_{20}$ | 88.86 | 26.23 | 66.41 | 99.19 | 23.19 | 57.81 | 74.82 | 20.17 | 64.45 | 71.59 | 15.37 | 50.00 | 76.08 | 15.10 | 42.58 | 91.28 | 20.75 | 61.72 |
| $I_{21}$ | 87.02 | 25.31 | 70.70 | 94.21 | 24.12 | 59.77 | 59.99 | 20.80 | 57.81 | 72.08 | 15.53 | 50.39 | 75.26 | 14.91 | 45.70 | 80.65 | 19.61 | 54.69 |
| $I_{22}$ | 82.08 | 16.74 | 51.17 | 102.12 | 25.81 | 61.72 | 72.84 | 21.61 | 62.11 | 49.79 | 11.83 | 32.03 | 79.31 | 14.14 | 46.88 | 86.05 | 16.39 | 42.97 |
| $I_{23}$ | 84.57 | 22.55 | 60.94 | 99.18 | 23.19 | 61.33 | 68.37 | 20.94 | 60.16 | 67.72 | 15.68 | 47.27 | 75.40 | 16.15 | 46.88 | 79.66 | 17.67 | 52.34 |
| $I_{24}$ | 90.38 | 28.01 | 74.22 | 103.79 | 22.92 | 63.28 | 131.89 | 18.08 | 59.38 | 65.69 | 14.82 | 43.75 | 108.95 | 15.32 | 52.73 | 93.00 | 20.48 | 59.77 |
| $I_{25}$ | 83.27 | 20.28 | 56.64 | 98.92 | 24.16 | 62.89 | 80.01 | 20.01 | 60.16 | 71.17 | 14.97 | 46.48 | 86.81 | 15.70 | 51.95 | 81.84 | 18.13 | 52.34 |

Table 6: The average length and CHAIR scores of descriptions generated by 12 MLLMs prompted by the 25 instructions on a subset of NoCaps containing 256 images.

| Metric | MiniGPT-4 BS | BS w/o HW | InstructBLIP BS | BS w/o HW | Lynx BS | BS w/o HW | LLaVA BS | BS w/o HW | Otter BS | BS w/o HW |
|---|---|---|---|---|---|---|---|---|---|---|
| $C_I \downarrow$ | 7.39 | 6.99 | 6.01 | 4.95 | 8.49 | 8.12 | 8.77 | 6.95 | 15.76 | 13.32 |
| $L_{C_I}(20) \downarrow$ | 5.33 | 5.20 | 2.35 | 2.24 | 3.26 | 3.69 | 7.22 | 5.67 | 8.76 | 7.89 |
| $L_{C_I}(40) \downarrow$ | 6.66 | 6.36 | 5.10 | 4.36 | 6.49 | 6.42 | 8.30 | 6.81 | 12.66 | 11.00 |
| $L_{C_I}(60) \downarrow$ | 7.98 | 7.53 | 7.86 | 6.47 | 9.72 | 9.16 | 9.38 | 7.96 | 16.56 | 14.10 |
| $L_{C_I}(80) \downarrow$ | 9.31 | 8.70 | 10.61 | 8.59 | 12.95 | 11.89 | 10.46 | 9.10 | 20.45 | 17.21 |
| $L_{C_I \text{GR}} \downarrow$ | 0.07 | 0.06 | 0.14 | 0.11 | 0.16 | 0.14 | 0.05 | 0.06 | 0.19 | 0.16 |
| $C_S \downarrow$ | 19.28 | 17.95 | 19.75 | 17.00 | 23.34 | 22.53 | 22.84 | 17.95 | 41.45 | 35.73 |
| $L_{C_S}(20) \downarrow$ | 9.27 | 8.98 | 5.61 | 5.33 | 8.00 | 8.87 | 14.48 | 10.32 | 15.31 | 14.26 |
| $L_{C_S}(40) \downarrow$ | 15.71 | 14.82 | 16.24 | 14.44 | 17.48 | 17.29 | 20.31 | 17.12 | 29.88 | 26.55 |
| $L_{C_S}(60) \downarrow$ | 22.15 | 20.65 | 26.87 | 23.56 | 26.97 | 25.71 | 26.14 | 23.92 | 44.45 | 38.84 |
| $L_{C_S}(80) \downarrow$ | 28.59 | 26.49 | 37.50 | 32.67 | 36.46 | 34.13 | 31.97 | 30.72 | 59.02 | 51.14 |
| $L_{C_S \text{GR}} \downarrow$ | 0.32 | 0.29 | 0.53 | 0.46 | 0.47 | 0.42 | 0.29 | 0.34 | 0.73 | 0.61 |
| T2I-CLIPRetrieval (R@1) $\uparrow$ | 38.76 | 38.18 | 29.66 | 30.20 | 39.76 | 40.18 | 35.02 | 33.52 | 21.72 | 22.08 |
| CLIPScore $\uparrow$ | 0.82 | 0.82 | 0.80 | 0.80 | 0.81 | 0.81 | 0.81 | 0.81 | 0.78 | 0.78 |
| RefCLIPScore $\uparrow$ | 0.81 | 0.81 | 0.83 | 0.83 | 0.80 | 0.80 | 0.82 | 0.82 | 0.79 | 0.79 |

| Metric | VPGTrans BS | BS w/o HW | LLaMA-Adapter-v2 BS | BS w/o HW | InternLM-XComposer BS | BS w/o HW | Qwen-VL BS | BS w/o HW | mPLUG-Owl2 BS | BS w/o HW |
|---|---|---|---|---|---|---|---|---|---|---|
| $C_I \downarrow$ | 7.28 | 6.84 | 11.91 | 11.48 | 7.54 | 6.73 | 6.39 | 5.63 | 9.08 | 7.55 |
| $L_{C_I}(20) \downarrow$ | 5.77 | 5.50 | 6.04 | 5.78 | 5.40 | 5.25 | 3.44 | 3.01 | 3.92 | 3.48 |
| $L_{C_I}(40) \downarrow$ | 6.87 | 6.48 | 9.29 | 8.97 | 7.82 | 6.96 | 5.36 | 4.77 | 7.39 | 6.36 |
| $L_{C_I}(60) \downarrow$ | 7.97 | 7.47 | 12.54 | 12.15 | 10.25 | 8.68 | 7.28 | 6.53 | 10.86 | 9.23 |
| $L_{C_I}(80) \downarrow$ | 9.08 | 8.45 | 15.80 | 15.34 | 12.67 | 10.40 | 9.20 | 8.29 | 14.33 | 12.11 |
| $L_{C_I \text{GR}} \downarrow$ | 0.06 | 0.05 | 0.16 | 0.16 | 0.12 | 0.09 | 0.10 | 0.09 | 0.17 | 0.14 |
| $C_S \downarrow$ | 17.19 | 16.64 | 32.39 | 31.33 | 18.03 | 16.67 | 20.20 | 18.15 | 28.20 | 23.78 |
| $L_{C_S}(20) \downarrow$ | 9.08 | 8.93 | 11.31 | 10.39 | 9.48 | 9.49 | 6.15 | 5.50 | 8.19 | 7.24 |
| $L_{C_S}(40) \downarrow$ | 15.01 | 14.61 | 22.99 | 22.09 | 19.18 | 17.78 | 15.31 | 14.00 | 21.66 | 18.95 |
| $L_{C_S}(60) \downarrow$ | 20.94 | 20.28 | 34.66 | 33.80 | 28.88 | 26.07 | 24.47 | 22.50 | 35.12 | 30.65 |
| $L_{C_S}(80) \downarrow$ | 26.86 | 25.96 | 46.34 | 45.50 | 38.58 | 34.36 | 33.63 | 31.00 | 48.59 | 42.36 |
| $L_{C_S \text{GR}} \downarrow$ | 0.30 | 0.28 | 0.58 | 0.59 | 0.48 | 0.41 | 0.46 | 0.43 | 0.67 | 0.59 |
| T2I-CLIPRetrieval (R@1) $\uparrow$ | 31.74 | 31.50 | 31.16 | 30.96 | 36.04 | 35.74 | 35.46 | 35.81 | 28.20 | 29.12 |
| CLIPScore $\uparrow$ | 0.80 | 0.79 | 0.81 | 0.81 | 0.81 | 0.81 | 0.82 | 0.82 | 0.81 | 0.81 |
| RefCLIPScore $\uparrow$ | 0.80 | 0.80 | 0.81 | 0.81 | 0.82 | 0.82 | 0.83 | 0.83 | 0.83 | 0.83 |

Table 7: Hallucination rate and quality of the generated image descriptions on MSCOCO after disabling hallucinogenic words. BS stands for Beam Search and HW stands for hallucinogenic Words.

| Metric | MiniGPT-4 BS | BS w/o HW | InstructBLIP BS | BS w/o HW | Lynx BS | BS w/o HW | LLaVA BS | BS w/o HW | Otter BS | BS w/o HW |
|---|---|---|---|---|---|---|---|---|---|---|
| $C_I \downarrow$ | 19.08 | 18.27 | 11.29 | 10.48 | 18.99 | 19.00 | 15.29 | 13.72 | 21.56 | 20.35 |
| $L_{C_I}(20) \downarrow$ | 14.53 | 14.08 | 6.52 | 6.44 | 13.79 | 14.53 | 12.68 | 11.51 | 15.49 | 15.16 |
| $L_{C_I}(40) \downarrow$ | 16.79 | 16.17 | 10.20 | 9.73 | 17.18 | 17.41 | 14.48 | 13.42 | 19.03 | 18.27 |
| $L_{C_I}(60) \downarrow$ | 19.05 | 18.26 | 13.88 | 13.03 | 20.57 | 20.29 | 16.29 | 15.33 | 22.58 | 21.38 |
| $L_{C_I}(80) \downarrow$ | 21.30 | 20.34 | 17.56 | 16.32 | 23.96 | 23.17 | 18.09 | 17.23 | 26.12 | 24.49 |
| $L_{C_I \text{GR}} \downarrow$ | 0.11 | 0.10 | 0.18 | 0.16 | 0.17 | 0.14 | 0.09 | 0.10 | 0.18 | 0.16 |
| $C_S \downarrow$ | 47.92 | 47.16 | 30.23 | 28.66 | 51.47 | 51.52 | 38.25 | 33.95 | 48.52 | 45.83 |
| $L_{C_S}(20) \downarrow$ | 23.75 | 23.53 | 13.33 | 13.04 | 36.07 | 35.98 | 24.15 | 20.90 | 25.38 | 25.22 |
| $L_{C_S}(40) \downarrow$ | 35.75 | 35.32 | 26.39 | 25.77 | 46.11 | 45.99 | 33.90 | 32.17 | 38.89 | 37.57 |
| $L_{C_S}(60) \downarrow$ | 47.76 | 47.10 | 39.45 | 38.51 | 56.16 | 56.00 | 43.66 | 43.44 | 52.40 | 49.93 |
| $L_{C_S}(80) \downarrow$ | 59.77 | 58.89 | 52.50 | 51.24 | 66.21 | 66.01 | 53.42 | 54.71 | 65.91 | 62.28 |
| $L_{C_S \text{GR}} \downarrow$ | 0.60 | 0.59 | 0.65 | 0.64 | 0.50 | 0.50 | 0.49 | 0.56 | 0.68 | 0.62 |
| T2I-CLIPRetrieval (R@1) $\uparrow$ | 50.34 | 50.72 | 47.48 | 48.12 | 55.24 | 53.94 | 47.38 | 46.08 | 31.46 | 30.94 |
| CLIPScore $\uparrow$ | 0.82 | 0.82 | 0.80 | 0.80 | 0.81 | 0.81 | 0.80 | 0.80 | 0.76 | 0.77 |
| RefCLIPScore $\uparrow$ | 0.82 | 0.82 | 0.82 | 0.83 | 0.81 | 0.81 | 0.81 | 0.82 | 0.78 | 0.79 |

| Metric | VPGTrans BS | BS w/o HW | LLaMA-Adapter-v2 BS | BS w/o HW | InternLM-XComposer BS | BS w/o HW | Qwen-VL BS | BS w/o HW | mPLUG-Owl2 BS | BS w/o HW |
|---|---|---|---|---|---|---|---|---|---|---|
| $C_I \downarrow$ | 15.07 | 14.80 | 18.83 | 18.21 | 12.32 | 11.76 | 11.01 | 10.29 | 11.01 | 10.47 |
| $L_{C_I}(20) \downarrow$ | 12.51 | 12.29 | 12.52 | 11.69 | 10.93 | 10.83 | 8.37 | 7.94 | 6.91 | 6.89 |
| $L_{C_I}(40) \downarrow$ | 14.39 | 14.15 | 16.07 | 15.41 | 12.74 | 12.08 | 10.69 | 10.03 | 10.82 | 10.56 |
| $L_{C_I}(60) \downarrow$ | 16.26 | 16.00 | 19.62 | 19.13 | 14.54 | 13.33 | 13.01 | 12.12 | 14.72 | 14.22 |
| $L_{C_I}(80) \downarrow$ | 18.14 | 17.85 | 23.17 | 22.85 | 16.34 | 14.58 | 15.33 | 14.21 | 18.63 | 17.89 |
| $L_{C_I \text{GR}} \downarrow$ | 0.09 | 0.09 | 0.18 | 0.19 | 0.09 | 0.06 | 0.12 | 0.10 | 0.20 | 0.18 |
| $C_S \downarrow$ | 36.16 | 35.70 | 45.31 | 43.83 | 28.64 | 27.91 | 26.50 | 24.75 | 26.14 | 24.56 |
| $L_{C_S}(20) \downarrow$ | 20.39 | 19.96 | 22.44 | 20.72 | 20.12 | 20.32 | 14.15 | 13.54 | 11.72 | 11.73 |
| $L_{C_S}(40) \downarrow$ | 31.95 | 31.60 | 35.31 | 33.90 | 31.22 | 30.50 | 25.00 | 23.50 | 25.45 | 24.88 |
| $L_{C_S}(60) \downarrow$ | 43.51 | 43.23 | 48.18 | 47.08 | 42.33 | 40.68 | 35.85 | 33.45 | 39.17 | 38.04 |
| $L_{C_S}(80) \downarrow$ | 55.07 | 54.87 | 61.04 | 60.26 | 53.44 | 50.87 | 46.71 | 43.40 | 52.90 | 51.20 |
| $L_{C_S \text{GR}} \downarrow$ | 0.58 | 0.58 | 0.64 | 0.66 | 0.56 | 0.51 | 0.54 | 0.50 | 0.69 | 0.66 |
| T2I-CLIPRetrieval (R@1) $\uparrow$ | 43.94 | 43.60 | 42.96 | 42.62 | 50.48 | 50.50 | 56.14 | 56.54 | 44.78 | 45.48 |
| CLIPScore $\uparrow$ | 0.79 | 0.79 | 0.80 | 0.80 | 0.79 | 0.79 | 0.82 | 0.82 | 0.81 | 0.81 |
| RefCLIPScore $\uparrow$ | 0.80 | 0.80 | 0.81 | 0.81 | 0.81 | 0.81 | 0.83 | 0.83 | 0.83 | 0.83 |

Table 8: Hallucination rate and quality of the generated image descriptions on NoCaps after disabling hallucinogenic words. BS stands for Beam Search and HW stands for hallucinogenic Words. Disabling hallucinogenic words can alleviate hallucinations in MLLMs.

