# OpenReview forum: "Toward a Stable, Fair, and Comprehensive Evaluation of Object Hallucination in Large Vision-Language Models"
_NeurIPS.cc/2024/Conference — NeurIPS 2024 poster_

### Official Review · Reviewer_onEV · 2024-07-07

**Soundness:** 3
**Presentation:** 3
**Contribution:** 4
**Rating:** 7
**Confidence:** 5

**Summary:**

This paper explores the stable evaluation of object hallucinations, which is a crucial challenge in large vision-language models. The authors provide the first systematic analysis of the underlying mechanism through which instructions affect hallucinations, based on comprehensive experiments. They report a linear correlation between the length of descriptions and the levels of object hallucinations. Furthermore, the authors propose a curve-based framework that incorporates description lengths to enable a stable evaluation of hallucinations. What I find particularly novel is that the slope of the curve is incorporated as a metric, which achieves a more comprehensive evaluation.

**Strengths:**

1. This work might provide valuable insights to the community. Firstly, while the impact of instructions on hallucinations is widely recognized, this work unveils a crucial aspect by demonstrating that instructions exert their influence through the modification of description lengths. This finding illuminates the previously unexplored mechanism underlying instruction-affected hallucinations. Secondly, they employ a curve-based evaluation method instead of relying solely on a single metric, which goes a new way in addressing hallucination evaluation. Thus, this work has the potential to inspire further research and exploration in hallucination evaluation.
2. The proposed curve-based hallucination evaluation method in this paper is intuitively reasonable, and the author provides substantial experimental evidence to support the motivation behind this method. The experimental results are clearly presented, and the corresponding analyses further enhance the persuasiveness of this work. Overall, the combination of the intuitive approach, extensive experiments, clear presentation of results, and insightful analyses makes this work convincing.

**Weaknesses:**

1. The proposed method realizes consistent evaluation by calculating the hallucination rate at a uniform length. However, the length distributions of descriptions generated by different LVLMs exhibit variations. In other words, some models tend to produce shorter descriptions while others generate longer ones. In light of this, I have concerns regarding the ability of this method to maintain its effectiveness under such circumstances.
2. In my view, the hallucination evaluation of a LVLM in practical requires a large instruction set that could simulate real-world applications of the LVLM. If the authors can build such a large instruction set as the benchmark, it would yield a significant contribution to the community.
3. The authors claim that their proposed evaluation method is fairer compared to other evaluation methods. However, the paper appears to lack experimental results to support this assertion.
4. The analysis of the stability of the “LeHaCE_GR” is lacking.
5. The selection of instructions may have a substantial impact on the fitted curve. It would be beneficial for the authors to provide further discussion on this aspect.

**Questions:**

1. Considering that shorter descriptions tend to have fewer hallucinations, have the authors explored the possibility of generating multiple concise descriptions with distinct focuses for the same image, and subsequently merging them into a comprehensive and detailed description?
2. What factors determine the slope of the length-hallucination curve for the model?
3. Since the authors introduce the slope of the length-hallucination curve as a valuable evaluation metric, it raises the question of what the intercept of the curve signifies. Is it feasible to incorporate the intercept into the evaluation framework?
4. Why does the average length of the image description generated by the Otter model, specifically under instruction I12, amount to only 2? Is there any misunderstanding here?

**Limitations:**

The paper briefly mentioned limitations.

---

> ### Author Rebuttal · Authors · 2024-08-06
>
> >**W1:** The proposed method realizes consistent evaluation by calculating the hallucination rate at a uniform length. However, the length distributions of descriptions generated by different LVLMs exhibit variations. In other words, some models tend to produce shorter descriptions while others generate longer ones. In light of this, I have concerns regarding the ability of this method to maintain its effectiveness under such circumstances.
>
> **Response to W1:** Thanks for your comment. In fact, calculating the hallucination rate at respective average lengths is a practice of the average-based framework, not our method. When the output lengths of different models are inconsistent, the average-based framework indeed suffers from evaluation inconsistency and unfairness. In our LeHaCE framework, by constructing the length-hallucination curve, we can evaluate the hallucination degree of different models at a specified uniform description length, thereby improving the consistency and fairness of the evaluation.
> ***
> >**W2:** In my view, the hallucination evaluation of a LVLM in practical requires a large instruction set that could simulate real-world applications of the LVLM. If the authors can build such a large instruction set as the benchmark, it would yield a significant contribution to the community.
>
> **Response to W2:** Thank you for your constructive suggestion. We will give it serious consideration.
> ***
>
> >**W3:** The authors claim that their proposed evaluation method is fairer compared to other evaluation methods. However, the paper appears to lack experimental results to support this assertion.
>
> **Response to W3:** Thanks for your comment. In Figure 4 of our paper, we present real data to visually demonstrate the fairness of our LeHaCE method. Our LeHaCE method constructs a length-hallucination curve to evaluate the hallucination levels of LVLMs at a uniform description length. This effectively mitigates the issue of length bias in hallucination level evaluation, providing a fairer evaluation.
> ***
> >**W4:** The analysis of the stability of the “LeHaCE_GR” is lacking.
>
> **Response to W4:** Following your suggestion, we supplement an experiment on the stability of LeHaCE_GR.The results are reported in Table 1 of author rebuttal pdf. From the results, we can observe that the consistency of LeHaCE_GR increases with the number of instructions.
> ***
> >**W5:** The selection of instructions may have a substantial impact on the fitted curve. It would be beneficial for the authors to provide further discussion on this aspect.
>
> **Response to W5:** Thanks for your valuable question. For the selection of instructions, a set of instructions that result in significant 5.differences in the model output length will aid in fitting the length illusion curve.
> ***
> >**Q1:** Considering that shorter descriptions tend to have fewer hallucinations, have the authors explored the possibility of generating multiple concise descriptions with distinct focuses for the same image, and subsequently merging them into a comprehensive and detailed description?
>
> **Response to Q1:** Thank you for your insightful suggestion. We have indeed tried this method. Specifically, we first had the LVLMs list the objects in the image, then prompted the LVLMs to generate descriptions for these objects individually, and finally summarized the information using the LVLM. However, due to the limited ability of LVLMs in listing objects and summarizing, this approach did not yield ideal results. Utilizing multimodal agent technology to achieve this idea with multiple different models is our future research direction.
> ***
> >**Q2:** What factors determine the slope of the length-hallucination curve for the model?
>
> **Response to Q2:** We believe that various factors influence the length-hallucination curve of LVLMs, such as the training data, the visual encoder and the language model in LVLMs.
> ***
> >**Q3:** Since the authors introduce the slope of the length-hallucination curve as a valuable evaluation metric, it raises the question of what the intercept of the curve signifies. Is it feasible to incorporate the intercept into the evaluation framework?
>
> **Response to Q3:** We believe that the intercept of the length-hallucination curve is not practically meaningful. This is because the output lengths of large vision-language models (LVLMs) are always positive, and for image descriptions, they typically contain at least a dozen words. Given this context, the intercept holds little practical significance.
> ***
> **Q4:** Why does the average length of the image description generated by the Otter model, specifically under instruction I12, amount to only 2? Is there any misunderstanding here?
>
> **Response to Q4:** This is because the Otter model always returns the names of the objects in the image rather than a description of the image when given the I12 instruction.

---

### Official Review · Reviewer_Pvzh · 2024-07-09

**Soundness:** 4
**Presentation:** 3
**Contribution:** 2
**Rating:** 5
**Confidence:** 4

**Summary:**

This work aims to establish a stable, fair, and comprehensive evaluation method for object hallucinations in large vision-language models. The authors discovered a positive correlation between the length of image descriptions and the degree of object hallucination. Building upon this observation, they developed a hallucination evaluation method named LeHaCE by fitting a length-hallucination curve. LeHaCE enables the evaluation at any given image description length, ensuring stability and fairness in the evaluation process. Additionally, LeHaCE involves the curve slope as a metric to evaluate the influence of image description length on the degree of object hallucination, thereby achieving a comprehensive evaluation. The motivation behind this work is reasonable, and the authors provide many experiments to support their claims. However, it is worth considering that the use of the linear fitting scheme, although straightforward, does somewhat diminish the novelty of the proposed method.

**Strengths:**

The experimental analysis conducted on instructions and hallucination is compelling and provides strong support for the main argument that the hallucination degree is positively correlated with the length of the description. While previous research (Yifan et al., 2023) has already shown the influence of instructions on hallucinations, this work takes it a step further by proposing that instructions indirectly influence hallucinations through the length of image descriptions. This sheds light on the reason behind the limitations of previous approaches that relied on average-based methods. Overall, this paper offers valuable insights into the evaluation of consistent hallucinations.

**Weaknesses:**

1. Although the rationale behind the length-hallucination curve is compelling, it is fitted using a relatively simplistic linear approach. Exploring more flexible and intricate fitting approaches is worth considering, as it has the potential to achieve higher fitting accuracy and more effective hallucination evaluation.
2. Since the proposed method relies on a fitted curve, it needs at least two instructions to evaluate LVLMs and cannot be used with just one instruction.The authors should discuss this limitation.
3. Lack of indepth discussion on the shortcomings of the proposed method. For instance, as shown in Table 2, why does LeHaCE exhibit poor stability on a few LVLMs when the number of instructions is three?
4. It seems that the selection of instructions might affect the stability of LeHaCE. It would be helpful to include more discussion on this aspect.
5. The current paper seems to have lots of results and experiments. As a reader, it is not very easy for me to get the main conclusion for each experiment. It would be good to highlight the conclusions so that the readers can understand the point easier.
6. Some typos need to be corrected: Line 79: lrv-instruction -> LRV-instruction. Line 92 Nope -> NOPE. Line 81 chatgpt -> ChatGPT. Table 2: Minigpt-4 -> MiniGPT-4.

**Questions:**

1. Does the complexity of the image content, such as the number of objects, influence the extent of hallucination in the model? It would be valuable to investigate additional factors that impact hallucination degrees.
2. Intuitively, the average-based framework can also be effective as long as there are enough instructions, such as 200 instructions. I'm wondering if this viewpoint is accurate?
3. Is the relative standard deviation an appropriate approach to evaluate stability, considering that stability in this context essentially refers to the consistency of multiple evaluation results?
4. Why does this work exclusively focus on object hallucinations? Is this a choice made by the authors or a limitation of the proposed method?
5. In Figure 5, why does LeHaCE show higher instability on LLaVA and Qwen-VL when the image description length is less than 20 words?

**Limitations:**

The paper includes a simple discussion of the limitations.

---

> ### Author Rebuttal · Authors · 2024-08-06
>
> > **W1:** Although the rationale behind the length-hallucination curve is compelling, it is fitted using a relatively simplistic linear approach. Exploring more flexible and intricate fitting approaches is worth considering, as it has the potential to achieve higher fitting accuracy and more effective hallucination evaluation.
>
> **Response to W1:** Following your suggestion, we supplement a comparative experiment for LeHaCE based on linear fitting versus quadratic and cubic polynomial fitting. The results are reported in Table 3 of the author rebuttal pdf.  These results show that linear fitting is significantly superior to polynomial fitting, especially when the number of instructions is small.
> ***
> >**W2:** Since the proposed method relies on a fitted curve, it needs at least two instructions to evaluate LVLMs and cannot be used with just one instruction. The authors should discuss this limitation.
>
> **Response to W2:** In the typical practice of evaluating hallucination levels in LVLMs, multiple instructions are usually used to enhance the stability of the evaluation results. Although LeHaCE cannot be used with just one instruction, this limitation does not affect its ability to provide stable evaluations. We will discuss this limitation in the final version.
> ***
>
> >**W3:** Lack of indepth discussion on the shortcomings of the proposed method. For instance, as shown in Table 2, why does LeHaCE exhibit poor stability on a few LVLMs when the number of instructions is three?
>
> **Response to W3:** Thank you for your valuable suggestions. LeHaCE exhibits instability with a low number of instructions because such a small sample is insufficient for an adequate fit of the length-hallucination curve. However, when the number of instructions is increased to five or more, LeHaCE consistently demonstrates improved stability.
> ***
>
> >**W4:** It seems that the selection of instructions might affect the stability of LeHaCE. It would be helpful to include more discussion on this aspect.
>
> **Response to W4:** For the selection of instructions, a set of instructions that result in significant differences in the model output length will aid in fitting the length illusion curve.
> ***
>
> >**W5:** The current paper seems to have lots of results and experiments. As a reader, it is not very easy for me to get the main conclusion for each experiment. It would be good to highlight the conclusions so that the readers can understand the point easier
>
> **Response to W5:** We will revise the paper to highlight the main conclusions of each experiment, making it easier for readers to grasp the key points.
> ***
>
> >**W6:** Some typos need to be corrected: Line 79: lrv-instruction -> LRV-instruction. Line 92 Nope -> NOPE. Line 81 chatgpt -> ChatGPT. Table 2: Minigpt-4 -> MiniGPT-4.
>
> **Response to W6:** Thanks for your careful review and we will revise these typos in our final version.
> ***
>
> >**Q1:** Does the complexity of the image content, such as the number of objects, influence the extent of hallucination in the model? It would be valuable to investigate additional factors that impact hallucination degrees.
>
> **Response to Q1:** Following your suggestion, We supplement an experiment about the correlation between the hallucination rates and the number of objects in the images for the Gemini, Llava, and MiniGPT-4 models on the COCO dataset, which were 0.08, -0.12, and 0.06, respectively. This indicates that the number of objects has almost no correlation with the hallucination degree of the models. Additionally, we found that LVLMs are more prone to hallucinations on some black-and-white images, suggesting that the style or domain of the images can affect the hallucination degree of LVLMs.
> ***
>
> >**Q2:** Intuitively, the average-based framework can also be effective as long as there are enough instructions, such as 200 instructions. I'm wondering if this viewpoint is accurate?
>
> **Response to Q2:** No. Considering that different LVLMs produce outputs of varying lengths for the same instructions, increasing the number of instructions still cannot alleviate the length bias of the average-based framework.
> ***
>
> >**Q3:** Is the relative standard deviation an appropriate approach to evaluate stability, considering that stability in this context essentially refers to the consistency of multiple evaluation results?
>
> **Response to Q3:** We believe that Relative Standard Deviation (RSD) is appropriate because the means of different metrics vary, making direct comparisons of standard deviations meaningless.Using RSD as an evaluation metric can eliminate the impact of the mean.
> ***
>
> >**Q4:** Why does this work exclusively focus on object hallucinations? Is this a choice made by the authors or a limitation of the proposed method?
>
> **Response to Q4:** Object hallucination is the most common type of hallucination in LVLMs. Following your suggestion, we evaluate the relation hallucinations and attribution hallucinations in image descriptions generated by Qwen-VL and InternLM-XComposer under 25 instructions. We conducted a model-based evaluation using Gemini-1.5-flash. The results are shown in Figure 2 of the Author Rebuttal attachment, from which we can observe that the degree of relation and attribution hallucination increases with the length of the descriptions. This demonstrates that relation hallucination and attribution hallucination are also influenced by length bias, suggesting that our findings and methods are applicable to these types of hallucinations as well. Expanding to other tasks will be the focus of our future work.
> ***
> >**Q5:** In Figure 5, why does LeHaCE show higher instability on LLaVA and Qwen-VL when the image description length is less than 20 words?
>
> **Response to Q5:** LLaVa and Qwen-VL have generally large output lengths, with minimums of 19 and 17, respectively (shown in Figure 3). This causes greater fitting deviations in LeHaCE's length-hallucination curve at shorter lengths, resulting in poorer consistency.

---

### Official Review · Reviewer_XTeq · 2024-07-11

**Soundness:** 2
**Presentation:** 3
**Contribution:** 2
**Rating:** 5
**Confidence:** 4

**Summary:**

The paper identifies a pitfall regarding the length of image descriptions in the current average-based LVLM hallucination evaluation framework. To address this, they propose a new Length-Hallucination Curve Based evaluation framework to enhance the fairness of evaluations. The paper observes that the degree of object hallucinations is primarily influenced by the length of image descriptions, with instructions indirectly affecting hallucinations through their impact on description lengths. They suggest using a linear regression curve for evaluation and develop two metrics based on this curve. Extensive experiments on multiple LVLMs with different instruction sets demonstrate the stability of their proposed new evaluation metrics.

**Strengths:**

- The observation is intuitive and validate by extensive experiments

- The paper is clearly written and easy to follow

- The evaluation is comprehensive in terms of numerous instructions and LVLMs

**Weaknesses:**

- Although paper observe the linear relation between the length of the image description and objection hallucination, there are still unanswered questions regarding the justification of the claim. Please see questions below.

- Some minor inconsistent typo, for example, the AEF and ABF in Figure 4.

- The evaluation only use CHAIR scores and scores of other aspects is not evaluated, for example, the detail or the coverage of the real objects in the description as in AMBER.

**Questions:**

- The paper grouped 25 different instructions to 5 instruction set. What’s the grouping strategy? How do the author group these instructions?

- The paper claimed that object hallucination is primarily influenced by the length of image descriptions, with instructions only indirectly affecting hallucinations through their effect on description lengths. How is this claim being validated? Specifically, how do the author validate that the length of the image description is the primary cause and is not also affecting the hallucinations indirectly through their effect on some hidden factors ? The observation could be due to the spurious correlation.

- Does the increased length of the image description also capture more real objects, or does it mainly consist of rephrasing and hallucinatory sentences?

**Limitations:**

Yes, the author adequately addressed the limitations and potential negative societal impact of their work.

---

> ### Author Rebuttal · Authors · 2024-08-06
>
> >**W1:** Although paper observe the linear relation between the length of the image description and objection hallucination, there are still unanswered questions regarding the justification of the claim. Please see questions below.
>
> >**Q2:** The paper claimed that object hallucination is primarily influenced by the length of image descriptions, with instructions only indirectly affecting hallucinations through their effect on description lengths. How is this claim being validated? Specifically, how do the author validate that the length of the image description is the primary cause and is not also affecting the hallucinations indirectly through their effect on some hidden factors ? The observation could be due to the spurious correlation.
>
> **Response to W1 and Q2:**  Thanks for your insightful comment. Intuitively, longer descriptions are more prone to hallucinations because the latter parts of the description can be influenced by the earlier parts, leading to cumulative hallucinations; Experimentally, we conducted extensive experiments across different models, datasets, and decoding strategies, and consistently observed a strong correlation between output length and the hallucination degree in these experiments. Theoretically establishing the causal relationship between output length and the hallucination degree will be a future direction of our work.
>
> ***
>
> >**W2**:Some minor inconsistent typo, for example, the AEF and ABF in Figure 4.
>
> **Response to W2:** Thanks for your careful review and we will revise these typos.
> ***
>
> >**W3**: The evaluation only use CHAIR scores and scores of other aspects is not evaluated, for example, the detail or the coverage of the real objects in the description as in AMBER.
>
> >**Q3:** Does the increased length of the image description also capture more real objects, or does it mainly consist of rephrasing and hallucinatory sentences?
>
> **Response to W3 and Q3:** Thanks for your valuable comment. The increased length of the image description also captures more real objects.  Following your suggestion, we analyzed the Coverage Ratio of image descriptions generated by 12 LVLMs under 25 instructions, defining the Coverage Ratio as |{real objects in description}|/|{all real objects in figure}|. The results are shown in Figure 1 of the Author Rebuttal attachment, from which we can observe that longer image descriptions result in a higher Coverage Ratio, capturing more real objects. This suggests that the Coverage Ratio is also influenced by length bias.
>
> Furthermore, we evaluated the stability of the Coverage Ratio when applying LeHaCE and the average-based framework. The results are in Table 2 of the author rebuttal pdf. From the results, we can observe that LeHaCE demonstrates greater stability compared to the average-based framework, and its stability further improves with the addition of more instructions.
> ***
>
> >**Q1:** The paper grouped 25 different instructions to 5 instruction set. What’s the grouping strategy? How do the author group these instructions?
>
> **Response to Q1:** Thanks for your comment. As described in Section 4.4, first paragraph: **“Specifically, LVLMs are prompted by three sets of different instructions to generate three sets of image descriptions. Each instruction set consists of multiple instructions randomly drawn from a pool of 25 instructions, with no overlap between instructions in different sets.”** We randomly selected three non-overlapping instruction sets from the 25 instructions to ensure reliable evaluation.
> ***

---

> > ### Comment · Reviewer_XTeq · 2024-08-13
> >
> > The additional experimental evaluations on coverage ratio and the possibilities of future work for the theoretical analysis on the relationship, which might be able to explain the phenomenon of the observed relationship, address my initial concerns. I have raised my overall score to 5 (borderline accept) accordingly.

---

> ### Author Response · Authors · 2024-08-14
>
> Thank you for taking the time and effort to evaluate our rebuttal and for adjusting the score.

---

### Official Review · Reviewer_ffbd · 2024-07-11

**Soundness:** 3
**Presentation:** 3
**Contribution:** 3
**Rating:** 7
**Confidence:** 5

**Summary:**

This work presents comprehensive experiments to study the relationship between description lengths and hallucinations in LVLMs. Based on the observed positive correlation, authors propose an approach of fitting a length-hallucination curve to evaluate object hallucinations. Speciffically, the curve allows for fair comparisons that are not influenced by varying lengths, through providing the hallucination degree corresponding to any given description length. Furthermore, the curve slope reflects the extent to which a LVLM's hallucination degree is affected by description lengths. The evaluation, considering both the value and slope, demonstrates stability and comprehensiveness, as supported by the conducted experiments. The authors' thorough and meticulous research on this issue is highly convincing, and the proposed method effectively showcases its effectiveness.

**Strengths:**

Hallucinations evaluation is a realistic and crucial task in the field of LVLMs, as hallucinations usually introduce misleading conclusions or even have disastrous outcomes. In this context, the authors perform a detailed experimental analysis on the impact of instructions on hallucinations, providing convincing evidence to support their motivation. Moreover, the proposed curve-based method is a simple yet effective approach, which is well-motivated by the observed linear correlation between description lengths and hallucination rates. The paper is well-written and effectively communicates its main contributions and techniques. Overall, the paper exhibits technical solidity.

**Weaknesses:**

1. The authors conduct experiments using only the beam search setting. Although I understand that beam search is widely used in hallucination evaluation of LVLMs/LLMs, it remains uncertain whether the observed correlation between the hallucination degree and the description length holds true under different decoding strategies. Thus, I recommend that the authors explore additional commonly used decoding strategies, such as greedy decoding, to provide a more comprehensive analysis.
2. The paper lacks a study about the influence of the instruction number on the length-hallucination curve. The fitted curve is directly affected by the number of samples, which corresponds to the number of instructions provided. It is therefore essential to thoroughly investigate the minimum number of instructions necessary for the proposed method.
3. The authors mention in the paper that the proposed method can "evaluate object hallucinations at any given image description length." In reality, when the given length deviates too much from the existing data, the fitting is likely to fail, leading to inaccurate results. The authors should use more cautious wording.
4. In my opinion, the impact of length might be mitigated by simply controlling the maximum generation lengths.The authors only mention this method in a footnote and believe it does not align with the actual usage scenarios of LVLMs. More in-depth discussions should be provided.
5. Some minor errors need to be corrected. For example, in line 42, "Figure 2&3" should be "Figures 2&3".
6. It appears inappropriate to represent a variable using only two letters. Consider replacing "hr" with "h_r".

**Questions:**

1. Why is the proposed method limited to large vision-language models? Could it be extended to large language models as well? It would be beneficial for the authors to provide a clear explanation or justification for this limitation.
2. Similarly, are the finding and method presented in this paper applicable to other forms of hallucination beyond object hallucinations, or other tasks, such as VQA?
3. What could potentially explain the phenomenon observed in Figure 2, where longer output lengths result in higher object hallucination degrees?
4. How are the 25 instructions used in experiments designed? Are they generated randomly or based on specific rules? Besides, why is it 25, and what difference would there be if there are more or less instructions?

**Limitations:**

The authors adequately addressed the limitations.

---

> ### Author Rebuttal · Authors · 2024-08-06
>
> > **W1**: The authors conduct experiments using only the beam search setting. Although I understand that beam search is widely used in hallucination evaluation of LVLMs/LLMs, it remains uncertain whether the observed correlation between the hallucination degree and the description length holds true under different decoding strategies. Thus, I recommend that the authors explore additional commonly used decoding strategies, such as greedy decoding, to provide a more comprehensive analysis.
>
>
> **Response to W1**
> Thanks for your valuable comment. Following your suggestion, we evaluate the hallucination degree of Qwen-VL and InternLM-XComposer under the greedy decoding strategy. The results are shown in Image 3 of the Author Rebuttal attachment, from which we can observe that the correlation between the hallucination degree and description length remains valid under the greedy decoding setting.
>
> Furthermore, we supplement a comparison of the average-based framework and LeHaCE under the greedy decoding setting. The results are reported in Table 4 of the Author Rebuttal attachment. The results show that LeHaCE's stability still surpasses that of the average-based method under the greedy decoding strategy.
>
> > **W2**: The paper lacks a study about the influence of the instruction number on the length-hallucination curve. The fitted curve is directly affected by the number of samples, which corresponds to the number of instructions provided. It is therefore essential to thoroughly investigate the minimum number of instructions necessary for the proposed method.
>
> **Response to W2:**
> Thanks for your comment. We analyze the impact of the number of instructions on the effectiveness of the LeHaCE framework in the second paragraph of Section 4.4: **In Table 2, we observe that when the number of instructions is very low, such as three, the stability of LeHaCE is compromised due to the difficulty in accurately fitting the length-hallucination curve. However, with just four or five instructions, LeHaCE consistently exhibits superior stability.** LeHaCE requires at least two instructions, but too few instructions can affect its consistency. When the number of instructions is greater than or equal to five, LeHaCE demonstrates excellent consistency.
>
> > **W3**: The authors mention in the paper that the proposed method can "evaluate object hallucinations at any given image description length." In reality, when the given length deviates too much from the existing data, the fitting is likely to fail, leading to inaccurate results. The authors should use more cautious wording.
>
> **Response to W3:**
> We will change the statement to "evaluate object hallucinations at any given image description length within a large range."
>
> >**W4**: In my opinion, the impact of length might be mitigated by simply controlling the maximum generation lengths. The authors only mention this method in a footnote and believe it does not align with the actual usage scenarios of LVLMs. More in-depth discussions should be provided.
>
> **Response to W4:**
> Thanks for your comment. Firstly, truncating the output to make the response length uniform does not reflect real usage scenarios, as users typically prefer complete responses. Secondly, truncating the output does not accurately control the description length, as some outputs may not reach the threshold length.
>
> >**W5**: Some minor errors need to be corrected. For example, in line 42, "Figure 2&3" should be "Figures 2&3".
>
> **Response to W5:**
> Thanks for your careful review and we will revise these typos in our final version.
>
> >**W6**: It appears inappropriate to represent a variable using only two letters. Consider replacing "hr" with "h_r".
>
> **Response to W6:**
> Thank you for your suggestion, we will revise it
>
> >**Q1**: Why is the proposed method limited to large vision-language models? Could it be extended to large language models as well? It would be beneficial for the authors to provide a clear explanation or justification for this limitation.
>
> **Response to Q1:**
> Thanks for this insightful question. The causes of hallucinations in LVLMs and LLMs share similarities, such as contradictions between the knowledge embedded in the parameters and the information in the context. Therefore, we believe that the proposed method has the potential to be extended to LLMs, which will be our future research direction.
>
> >**Q2**: Similarly, are the finding and method presented in this paper applicable to other forms of hallucination beyond object hallucinations, or other tasks, such as VQA?
>
> **Response to Q2:**
> Following your suggestion, we evaluate the relation hallucinations and attribution hallucinations in image descriptions generated by Qwen-VL and InternLM-XComposer under 25 instructions. We conducted a model-based evaluation using Gemini-1.5-flash. The results are shown in Figure 2 of the Author Rebuttal attachment, from which we can observe that the degree of relation and attribution hallucination increases with the length of the descriptions, indicating that our findings and method are applicable to other forms of hallucination. Expanding to other tasks will be the focus of our future work.
>
> >**Q3**: What could potentially explain the phenomenon observed in Figure 2, where longer output lengths result in higher object hallucination degrees?
>
> **Response to Q3:**
> In Appendix 6.2, we further explore this phenomenon and find that MLLMs are more likely to employ hallucinogenic words in generating lengthy and detailed image descriptions, resulting in a higher hallucination rate.
>
> > **Q4**: How are the 25 instructions used in experiments designed? Are they generated randomly or based on specific rules? Besides, why is it 25, and what difference would there be if there are more or less instructions?
>
> **Response to Q4:**
> To obtain a diverse and extensive set of instructions, we referred to those from existing works and additionally designed some of our own. The value of 25 was not meticulously chosen.

---

### Author Rebuttal · Authors · 2024-08-06

We thank all the reviewers and area chairs for your time and effort during the review process. We are encouraged to hear that our work has **clear and well-written presentations** (by all Reviewers), **good motivation** (by Reviewer Pvzh and Ffbd), **convincing analysis** (by Reviewer Pvzh and onEV), **novel** (by Reviewer onEV) **and effective** (by Reviewer Ffbd) **technical contributions**,  **valuable insights** (by ReviewerPvzh and onEV), **extensive experiments** (by Reviewer Pvzh and onEV), and **comprehensive evaluation** (by reviewer XTeq ).

During the rebuttal phase, we meticulously give point-by-point responses to your comments, and further add the additional experiments and figures into the one-page supplementary PDF. Especially,
- we provided extensive evaluations of LeHaCE on more hallucination evaluation metrics, including coverage ratio, relation hallucination, attribute hallucination, and more decoding strategies (greedy decoding), which further validate the effectiveness and versatility of our findings and method.
- We also provided comprehensive ablation experiments on the fitting methods for LeHaCE, which validated the reasonableness of using linear fitting.
- Furthermore, we conducted a more comprehensive and in-depth analysis of our findings.

We hope that our responses adequately address all your concerns and meet the expectations of the conference committee.

---

### Decision · Program_Chairs · 2024-09-25

**Decision:**

Accept (poster)

**Comment:**

This paper finds that the description length strongly correlates with the levels of object hallucinations. Authors further propose to use the slope of the curve as a metric for evaluation. Reviewers consistently gave accept, borderline accept, borderline accept, and accept. Reviewers generally like this paper and think the papeer is novel and easy to understand. The AC agrees on these aspects and recommend accept.